# A PROBABLISTIC AUTOMATA LEARNING APPROACH FOR ANALYZING AND SAMPLING CONSTRAINED LLM

## ABSTRACT

We define a congruence that copes with null next-symbol probabilities that arise when the output of a language model is constrained by some means during text generation. We develop an algorithm for efficiently learning the quotient with respect to this congruence and evaluate it on case studies for analyzing statistical properties of LLM.

## 1 INTRODUCTION

Many works have studied neural language models, such as Recurrent Neural Networks (RNN) and Transformers, through the analysis of surrogate automata of different sorts obtained from the former in a variety of ways, with the purpose of verifying or explaining their behavior Wang et al. (2018); Weiss et al. (2018); Khmelnitsky et al. (2021); Mayr et al. (2023); Muškardin et al. (2023).

Recently, several papers proposed to analyze neural sequence-processing models by composing them with automata or regular expressions in order to verify properties on-the-fly while learning Mayr et al. (2021), assess the existence of memorization, bias, or toxicity Kuchnik et al. (2023), and guide text generation Willard & Louf (2023). However, they have not been applied to language models, but language recognizers, this is the case of Mayr et al. (2021), or they lack formalization.

An important problem that arises when synchronizing a neural language model with a guiding automaton or constraining text generation with common sampling strategies, such as top-$k$, is the occurrence of symbols with null probabilities. A consequence of this, for instance, is that generation may not terminate. Moreover, this implies the model does not define a probability distribution over finite strings.

The contribution of the paper is threefold: 1) the definition of a Myhill-Nerode-like congruence over strings which takes into account the occurrence of zero-probabilities, that provides an underlying formal basis for learning of probabilistic deterministic finite automata (PDFA) Vidal et al. (2005) from neural language models constrained by automata and sampling strategies; 2) the development of the **Omit-Zero** algorithm for learning the quotient with respect to this congruence, which shows to be more efficient than other algorithms for the experiments carried out; 3) a framework for analyzing statistical properties of LLM based on the previous two.

In Sec. 2, we address the question of dealing with null next-symbol probabilities that appear when constraining the output of a language model by composing it with an automaton and/or a sampling strategy, such as the top $k$ most likely symbols. We do this by defining an appropriate congruence that induces a quotient PDFA without zero-probability transitions. In Sec. 3, we adapt the learning algorithm of Mayr et al. (2023) to efficiently learn the quotient PDFA. In Sec. 4, we discuss issues that arise when analyzing real large language models, in particular the role of tokenizers, and apply the algorithm on problems discussed in Kuchnik et al. (2023); Willard & Louf (2023) when generating text with GPT2. Experimental results show the interest of our approach.

## 2 LANGUAGE MODELS

Let $\Sigma$ be a finite set of *symbols*, $\Sigma^*$ the set of finite *strings*, $\lambda \in \Sigma^*$ the *empty* string, and $\Sigma_{\$} \triangleq \Sigma \cup \{\$\}$, where $\$ \notin \Sigma$ is a special symbol used to denote *termination*. We denote $\Delta(\Sigma_{\$})$ the

*probability simplex* over $\Sigma_\$$, that is, the set of all $\rho : \Sigma_\$ \to \mathbb{R}_+$ such that $\sum_{\sigma \in \Sigma_\$} \rho(\sigma) = 1$. The *support* of $\rho \in \Delta(\Sigma_\$)$ is $\mathsf{supp}(\rho) \triangleq \{\sigma \in \Sigma_\$ \mid \rho(\sigma) > 0\}$.

**Definition 1.** *A* language model *is a total function* $\mathcal{L} : \Sigma^* \to \Delta(\Sigma_\$)$.

Def. 1 abstracts away from particular computational mechanisms used to implement concrete language models such as neural models, for example, RNN and Transformers, or state-transition models, for instance, Markov chains or PDFA. This work leverages PDFA as a foundation for analyzing neural models. Moreover, PDFA offer a simple and intuitive formalism, along with graphical representations, to illustrate examples of language models. To this end, we provide their definition here.

Following Mayr et al. (2023), a PDFA $\mathcal{A}$ over $\Sigma$ as a tuple $(Q, q_{\mathrm{in}}, \pi, \tau)$, where $Q$ is a finite set of states, $q_{\mathrm{in}} \in Q$ is the initial state, $\pi : Q \to \Delta(\Sigma_\$)$, and $\tau : Q \times \Sigma \to Q$. Both $\pi$ and $\tau$ are total functions. The extensions $\tau^*$ and $\pi^*$ are defined as follows: $\tau^*(q, \lambda) \triangleq q$ and $\tau^*(q, \sigma u) \triangleq \tau^*(\tau(q, \sigma), u)$, and $\pi^*(q, u) \triangleq \pi(\tau^*(q, u))$. When $q = q_{\mathrm{in}}$, we omit the state $q$ in the notation above and simply write $\tau^*(u)$ and $\pi^*(u)$. $\mathcal{A}$ defines the language model such that $\mathcal{A}(u) \triangleq \pi^*(u)$. Fig. 1 gives examples of PDFA. The number below $q$ is the probability of termination $\pi(q)(\$)$, and the one associated with an outgoing transition labeled $\sigma$ corresponds to $\pi(q)(\sigma)$.

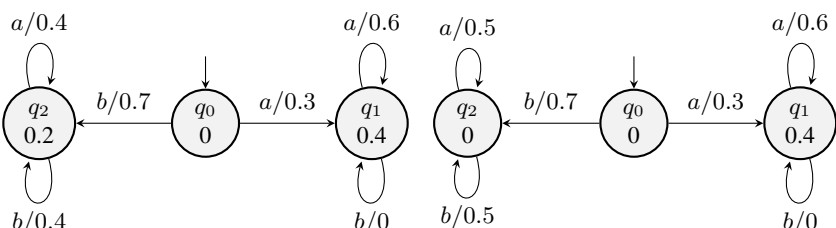

Figure 1: PDFA $\mathcal{A}$ (left) and $\mathcal{B}$ (right) over $\Sigma = \{a, b\}$ with $q_{\mathrm{in}} = q_0$.

**Sampling** $\mathcal{L}$ can be used to generate random strings $x \in \Sigma^*$ with $x_i \sim \mathcal{L}(x_{<i})$, for $i \geq 1$, where $x_i$ is the $i$-th symbol and $x_{<i} = x_1 \ldots x_{i-1}$ with $x_{<1} \triangleq \lambda$. That is, by sampling the next symbol to concatenate from the distribution of the prefix until the termination symbol is selected.

In general, this procedure may not terminate. In fact, $\mathcal{L}$ uniquely defines a probability distribution over $\Sigma^* \cup \Sigma^\omega$, where $\Sigma^\omega$ denotes the set of all infinite strings. More precisely, if we let $P : \Sigma^* \to \mathbb{R}_+$ to be defined recursively as

$$P(u\sigma) \triangleq P(u) \cdot \mathcal{L}(u)(\sigma), \quad P(\lambda) \triangleq 1,$$

and $P_\$ : \Sigma^* \to \mathbb{R}_+$ to be defined by $P_\$(u) \triangleq P(u) \cdot \mathcal{L}(u)(\$)$, thenthere exists a unique probability distribution $\boldsymbol{P}$ over $\Sigma^* \cup \Sigma^\omega$ whose prefix probabilities are given by $P$ and whose restriction to $\Sigma^*$ is given by $P_\$$:

**Proposition 2.1.** *Let* $\mathcal{L} : \Sigma^* \to \Delta(\Sigma_\$)$ *be a language model. There exists a unique Borel*[1] *probability measure* $\boldsymbol{P}$ *in* $\Sigma^* \cup \Sigma^\omega$ *such that*

$$P(w) = \boldsymbol{P}\big\{x \in \Sigma^* \cup \Sigma^\omega : w \in \mathsf{pref}(x)\big\} \text{ and } P_\$(w) = \boldsymbol{P}\big\{w\big\}$$

*for all* $w \in \Sigma^*$. *Here* $\mathsf{pref}(x)$ *denotes the set of all prefixes in* $\Sigma^*$ *of* $x$, *including* $\lambda$ *and* $x$ *itself*.

*Proof.* See Appendix A. $\qquad\square$

In general, the probability $\boldsymbol{P}$ provided by Prop. 2.1 does not concentrate its mass on $\Sigma^*$. Consequently, $P_\$$ is not a proper probability distribution over $\Sigma^*$, as it may not sum to 1 Vidal et al. (2005). In such cases, there is a positive probability that the sampling procedure described above will fail to terminate. Necessary and sufficient conditions for termination involve properties of the probabilities associated with the terminal symbol Du et al. (2023). For example, in the case of a PDFA, the

---

[1]Borel means here that the measure $\boldsymbol{P}$ is defined over the $\sigma$-algebra generated by the cylinder sets. See Appendix A for more details.

sampling procedure terminates if, for every state $q$, there exists a reachable state $q'$ (via transitions from $q$) where the terminal symbol appears with positive probability. As an example consider $\mathcal{A}$ in Fig. 1. Even though $\pi_{\mathcal{A}}(q_0)(\$) = 0$, we have that $P_\$$ defines a probability distribution in $\Sigma^*$ since $\sum_{u \in \Sigma^*} P_\$(u) = 0.3 \cdot 0.4 \sum_{n=0}^{\infty} 0.6^n + 0.7 \cdot 0.2 \sum_{n=0}^{\infty} 0.8^n = 0.3 + 0.7 = 1$. However, this is not the case for $\mathcal{B}$, with $\pi_{\mathcal{A}}(q_2)(\$) = 0$, since in this case $\sum_{u \in a\Sigma^*} P_\$(u) + \sum_{u \in b\Sigma^*} P_\$(u) = 0.3 \cdot 0.4 \sum_{n=0}^{\infty} 0.6^n = 0.3 < 1$. $\mathcal{B}$ can actually be obtained from $\mathcal{A}$ by constraining the set of symbols to sample from to the top-2 most likely ones: $\mathsf{top}_2(\pi_{\mathcal{A}}(q_2)) = \{a, b\}$, and normalizing the probabilities. It results in that no finite string starting with symbol $b$ can be sampled in $\mathcal{B}$ with distribution $P_\$$.

Using $\mathsf{top}_r$ or $\mathsf{top}_p$ (most likely symbols with a cumulative probability cutoff of $p$) is usual practice when sampling from an LLM. Since this may induce non-termination at the time of generating strings, it is relevant to formalize the effect of these constraints on $\mathcal{L}$.

A *sampling strategy* is a map $\mathsf{samp} : \Delta(\Sigma_\$) \to \Delta(\Sigma_\$)$ is such that $\mathsf{supp}(\mathsf{samp}(\rho)) \subseteq \mathsf{supp}(\rho)$ for all $\rho \in \Delta(\Sigma_\$)$. We denote $\mathsf{samp}(\mathcal{L})$ the language model obtained by applying $\mathsf{samp}$ to $\mathcal{L}(u)$ for all $u \in \Sigma^*$. For example, in Fig. 1, $\mathcal{B} = \mathsf{samptop}_2(\mathcal{A})$, where:

$$\mathsf{samptop}_r(\rho)(\sigma) = \begin{cases} \frac{\rho(\sigma)}{\sum_{\sigma' \in \mathsf{top}_r(\rho)} \rho(\sigma')} & \text{if } \sigma \in \mathsf{top}_r(\rho) \\ 0 & \text{otherwise} \end{cases} \tag{1}$$

**Congruences** $P$ is used in Carrasco & Oncina (1999); Vidal et al. (2005) to define the following equivalence relation $\equiv$ on $\Sigma^*$ which is a *congruence* with respect concatenating a symbol:

$$u \equiv v \iff^{\triangle} \forall w \in \Sigma^*. \frac{P(uw)}{P(u)} = \frac{P(vw)}{P(v)} \tag{2}$$

Notice that zero probabilities in the denominator give undefined quotients. In the case one side of (2) is undefined, the equality must be understood as implying that the other side is also undefined.

We define $\mathbb{1}_{\mathcal{L}} : \Sigma^* \to \{0, 1\}$ such that $\mathbb{1}_{\mathcal{L}}(u) = 1$ iff $P(u) > 0$.

**Proposition 2.2.** *For all $u, v \in \Sigma^*$. $u \equiv v$ if and only if*

$$\mathbb{1}_{\mathcal{L}}(u) = \mathbb{1}_{\mathcal{L}}(v) \text{ and } \forall w \in \Sigma^*. \mathbb{1}_{\mathcal{L}}(uw) = \mathbb{1}_{\mathcal{L}}(vw) = 1 \implies \mathcal{L}(uw) = \mathcal{L}(vw). \tag{3}$$

*Proof.* See Appendix B. $\square$

Resorting to some kind of tolerance relation between distributions is usual practice when it comes to approximating the behavior of language models with probabilistic automata in order to group in a single state strings which continuations slightly differ in probability. For instance, in Weiss et al. (2019); Clark & Thollard (2004), two distributions are considered similar if their *variation distance*

$$\boldsymbol{d}(\rho, \rho') \triangleq \max_{\sigma \in \Sigma_\$} |\rho(\sigma) - \rho'(\sigma)|$$

is less than or equal to a specified tolerance threshold $t$. Eventually, this grouping could result in an approximation with a finite number of states even if the image of the language model contains infinitely many distributions, while keeping the error of the approximation as small as desired or preserving the property to be checked.

However, this approach has a significant limitation: the induced relation on $\Delta(\Sigma_\$)$ is not transitive, and thus, it cannot be extended to a congruence relation on $\Sigma^*$. To overcome this issue, we propose using equivalence relations instead. This leads to a well-defined notion of algebraic quotient and allows capturing the behavior of the language model under usual sampling strategies such as (1). Several equivalence relations are of interest, some examples having been employed in Mayr et al. (2023):

**Quantization** Given a *quantization parameter* $\kappa \in \mathbb{N}$, $\kappa \geq 1$, the quantization partition of the interval $[0, 1]$ is defined as $\left\{ [0], \left(0, \kappa^{-1}\right), \left[\kappa^{-1}, 2\kappa^{-1}\right), \ldots, \left[(\kappa - 1)\kappa^{-1}, 1\right), [1] \right\}$. For $\rho, \rho' \in \Delta(\Sigma_\$)$, we define $\rho =_\kappa \rho'$ if and only if for each symbol $\sigma$, $\rho(\sigma)$ and $\rho'(\sigma)$ belong to the same quantization interval. Notice that $\rho =_\kappa \rho'$ implies $\boldsymbol{d}(\rho, \rho') \leq 1/\kappa$.

3

**Top** For $r \in \mathbb{N}$ and $\rho, \rho' \in \Delta(\Sigma_\$)$, we define $\rho =_{\text{top}_r} \rho'$ if and only if $\rho$ and $\rho'$ share the same support and $\text{top}_r(\rho) = \text{top}_r(\rho')$. A finer relation can be defined by looking at their ranking.

Let $E$ be an equivalence relation in $\Delta(\Sigma_\$)$. We denote $\rho =_E \rho'$ the equivalence, $[\Delta(\Sigma_\$)]_E$ and $[\rho]_E$ the quotient of $\Delta(\Sigma_\$)$ and the class of $\rho$ induced by $E$ respectively. We require:

$$\text{supp}(\rho) = \text{supp}(\rho') \text{ whenever } \rho =_E \rho' \tag{4}$$

Motivated by (3) we generalize (2) as follows:

**Definition 2.** *For $u, v \in \Sigma^*$, $u \equiv_E v$ if and only if*

$$\mathbb{1}_{\mathcal{L}}(u) = \mathbb{1}_{\mathcal{L}}(v) \text{ and } \forall w \in \Sigma^*. \ \mathbb{1}_{\mathcal{L}}(uw) = \mathbb{1}_{\mathcal{L}}(vw) = 1 \implies \mathcal{L}(uw) =_E \mathcal{L}(vw). \tag{5}$$

We denote $[\![\Sigma^*]\!]_E$ the set of equivalence classes of $\equiv_E$ and $[\![u]\!]_E$ the class of $u$. Since $\mathbb{1}_{\mathcal{L}}(u) = \mathbb{1}_{\mathcal{L}}(v)$ for all $u \equiv_E v$, we extend $\mathbb{1}_{\mathcal{L}}$ to $[\![\Sigma^*]\!]_E$ and write $\mathbb{1}_{\mathcal{L}}([\![u]\!])$.

**Proposition 2.3.** $\equiv_E$ *is a congruence:* $\forall u, v \in \Sigma^*. \ u \equiv_E v \implies \forall \sigma \in \Sigma. \ u\sigma \equiv_E v\sigma.$

*Proof.* Let $u \equiv_E v$. If $\mathbb{1}_{\mathcal{L}}(u) = \mathbb{1}_{\mathcal{L}}(v) = 0$, then $\mathbb{1}_{\mathcal{L}}(uw) = \mathbb{1}_{\mathcal{L}}(vw) = 0$ for all $w \in \Sigma^*$. Then $u\sigma \equiv_E v\sigma$ trivially.

Suppose now that $\mathbb{1}_{\mathcal{L}}(u) = \mathbb{1}_{\mathcal{L}}(v) = 1$ and let $\sigma \in \Sigma$. We have $\mathbb{1}_{\mathcal{L}}(u\sigma) = \mathbb{1}_{\mathcal{L}}(v\sigma)$ by Req. 4. Let $w \in \Sigma^*$ be arbitrary, since concatenation of strings is associative, if $\mathbb{1}_{\mathcal{L}}((u\sigma)w) = \mathbb{1}_{\mathcal{L}}((v\sigma)w) = 1$, then $\mathbb{1}_{\mathcal{L}}(u(\sigma w)) = \mathbb{1}_{\mathcal{L}}(v(\sigma w)) = 1$ and by assumption $\mathcal{L}(u(\sigma w)) =_E \mathcal{L}(v(\sigma w))$. Thus $\mathcal{L}((u\sigma)w) =_E \mathcal{L}((v\sigma)w)$. This proves that $u\sigma \equiv_E v\sigma$. $\qquad\square$

Let $\equiv_E^\bullet$ be the congruence in $\Sigma^*$ defined in Mayr et al. (2023):

$$u \equiv_E^\bullet v \ \overset{\Delta}{\iff} \ \forall w \in \Sigma^*. \ \mathcal{L}(uw) =_E \mathcal{L}(vw) \tag{6}$$

We denote by $\mathbf{0}$ the $\equiv_E$-class of all $u \in \Sigma^*$ with $\mathbb{1}_{\mathcal{L}}(u) = 0$.

**Proposition 2.4.** *There exists a one-to-one map $\phi : [\![\Sigma^*]\!]_E \setminus \{\mathbf{0}\} \to [\![\Sigma^*]\!]_E^\bullet$.*

*Proof.* Let $\alpha : [\![\Sigma^*]\!]_E \setminus \{\mathbf{0}\} \to \Sigma^*$ be any function satisfying $\alpha(c) \in c$ for all $c \in [\![\Sigma^*]\!]_E \setminus \{\mathbf{0}\}$. In other words, $\{\alpha(c) : c \in [\![\Sigma^*]\!]_E \setminus \{\mathbf{0}\}\}$ is a set of representatives of the classes. Let $\beta : \Sigma^* \to [\![\Sigma^*]\!]_E^\bullet$ be the quotient map $\beta(u) = [\![u]\!]_E^\bullet$. Define $\phi = \beta \circ \alpha$.

Let $c, c' \in [\![\Sigma^*]\!]_E \setminus \mathbf{0}$ be such that $\phi(c) = \phi(c')$. Denote $u = \alpha(c)$ and $v = \alpha(c')$. By construction $\mathbb{1}_{\mathcal{L}}(u) = \mathbb{1}_{\mathcal{L}}(v) = 1$ and by Def. (6) we have $\mathcal{L}(uw) =_E \mathcal{L}(vw)$ for all $w \in \Sigma^*$. In particular $u \equiv_E v$, or equivalently $c = [\![u]\!]_E = [\![v]\!]_E = c'$. $\qquad\square$

**Corollary 2.1.** *If $[\![\Sigma^*]\!]_E^\bullet$ is finite then $[\![\Sigma^*]\!]_E$ is finite, and $\#[\![\Sigma^*]\!]_E \leq \#[\![\Sigma^*]\!]_E^\bullet + 1$.*

For PDFA, $\equiv_E$ (similarly for $\equiv_E^\bullet$) can be rephrased over $Q$ as follows: $\forall u, v \in \Sigma^*$

$$\tau^*(u) \equiv_E \tau^*(v) \ \overset{\Delta}{\iff} \ u \equiv_E v \tag{7}$$

Fig. 2(left) illustrates the difference between $\equiv_E$ and $\equiv_E^\bullet$. $E$ is equality. States $q_0, q_1$, and $q_2$ are not $\equiv_E^\bullet$-equivalent: $\pi(q_2) \neq \pi(q_0) = \pi(q_1)$, and $\pi^*(q_0, b) \neq \pi^*(q_1, b)$. However, $q_0 \equiv_E q_1$ because $\mathbb{1}(u) = 1$ and $\pi^*(q_0, u) = \pi^*(q_1, u)$, for $u \in \{a\}^*$, and $\mathbb{1}(u) = 0$, for $u \in b\Sigma^*$.

**Proposition 2.5.** *Let $\mathcal{L} : \Sigma^* \to \Delta(\Sigma_\$)$, $u, v \in \Sigma^*$ such that $\mathbb{1}_{\mathcal{L}}(u) = \mathbb{1}_{\mathcal{L}}(v) = 1$. For every $w \in \Sigma^*$ such that $\mathbb{1}_{\mathcal{L}}(uw) = 1$, if $\mathcal{L}(uw) \neq_E \mathcal{L}(vw)$, then there exists $w' \in \text{pref}(w)$ such that $\mathcal{L}(uw') \neq_E \mathcal{L}(vw')$, and $\mathbb{1}_{\mathcal{L}}(vw') = 1$.*

*Proof.* If $\mathbb{1}_{\mathcal{L}}(vw) = 1$ then $w' = w$. Otherwise, there exists $w'\sigma \in \text{pref}(w)$ such that $1 = \mathbb{1}_{\mathcal{L}}(vw') \neq \mathbb{1}_{\mathcal{L}}(vw'\sigma) = 0$. Hence, $\text{supp}(\mathcal{L}(uw')) \neq \text{supp}(\mathcal{L}(vw'))$ because $\mathbb{1}_{\mathcal{L}}(uw'\sigma) = 1$. Thus, by Req. 4, $\mathcal{L}(uw') \neq_E \mathcal{L}(vw')$. $\qquad\square$

For the sake of readability, we assume hereinafter that, unless stated otherwise, the congruence relation is associated with an equivalence $E$ and omit the subscript.

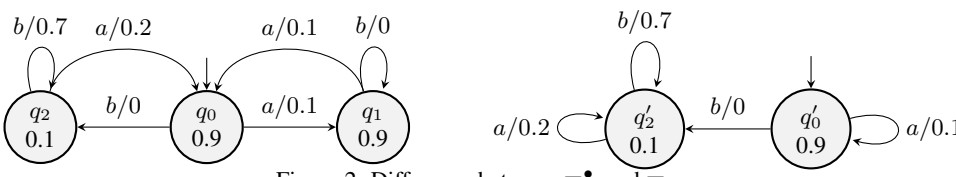

Figure 2: Difference between $\equiv_E^\bullet$ and $\equiv_E$.

**Quotients** $\equiv$ induces a *quotient* $\overline{\mathcal{L}} : [\![\Sigma^*]\!] \to [\Delta(\Sigma_{\$})]$ defined as follows: $\overline{\mathcal{L}}([\![u]\!]) \triangleq [\mathcal{L}(u)]$. For a PDFA $\mathcal{A}$, its quotient $\overline{\mathcal{A}}$ is $(\overline{Q}, \overline{q}_{\text{in}}, \overline{\pi}, \overline{\tau})$, where $\overline{Q} \triangleq [\![reach(Q)]\!]$, with $reach(Q) \triangleq \bigcup_{u \in \Sigma^*} \tau^*(u)$, $\overline{q}_{\text{in}} \triangleq [\![q_{\text{in}}]\!]$, $\overline{\pi}([\![q]\!]) \triangleq [\pi(q)]$, and $\overline{\tau}([\![q]\!], \sigma) \triangleq [\![\tau(q, \sigma)]\!]$ for all $\sigma \in \Sigma$.

From (7), it follows that each $\overline{q} \in \overline{Q}$ can be represented by an *access* string $u$ with $\overline{q} = [\![\tau^*(u)]\!]$. Let $\alpha(\overline{q})$ be the designated access string of $\overline{q}$. W.l.o.g., $\alpha(\overline{q}_{\text{in}}) \triangleq \lambda$. Given $\overline{\mathcal{A}}$, we can construct a PDFA $\overline{\mathcal{A}}_\alpha \triangleq (\overline{Q}, \overline{q}_{\text{in}}, \pi_\alpha, \overline{\tau})$, where for all $\overline{q} \in \overline{Q}$, $\overline{\pi}_\alpha(\overline{q}) \triangleq \pi^*(\alpha(\overline{q}))$. Clearly, all choices of $\alpha$ yield isomorphic PDFA that are $\equiv$-equivalent. Thus, unless necessary, we omit $\alpha$ and use $\overline{\mathcal{A}}$ to refer to any such PDFA. $\overline{\mathcal{A}}$ is the smallest PDFA which is $\equiv$-equivalent to $\mathcal{A}$. As an example, let $\mathcal{A}$ and $\mathcal{B}$ be the PDFA in Fig. 2(left) and (right), respectively. Since all states of $\overline{\mathcal{A}}$ are $\not\equiv^\bullet$, we have that $\overline{\mathcal{A}}_{\equiv^\bullet} = \mathcal{A}$. However, $\overline{\mathcal{A}}_{\equiv} = \mathcal{B}$ because $q_0 \equiv q_1 \not\equiv q_2$.

Here, it is worth to mention that while the choice of $\alpha$ is irrelevant with respect to the congruence, different ones may result in different $P_{\$}$. Nevertheless, if $E$ induces convex classes, as is the case for quantization, rank, and top defined in Mayr et al. (2023), it is always possible to define $\overline{\pi}(\overline{q})$ as a convex combination of distributions in $[\overline{\pi}_\alpha(\overline{q})]_E$.

# 3 LEARNING ALGORITHM

Based on the results of Sec. 2, we developed the algorithm **Omit-Zero**, to learn $\equiv$-minimal PDFA. It is a variant of QNT (Mayr et al. (2023)) that differs in specific steps indicated with boxes in Alg. 1. **Omit-zero** maintains a tree $T$ whose nodes are strings which are partitioned in two sets: $Acc \subset \Sigma^*$ and $Dis \subset \Sigma^*$ of *access* and *distinguishing* strings, respectively. $Acc$ is the set of *leafs*, representing congruence classes. Each $u \in Acc$ is labelled with the distribution $\mathcal{L}(u)$. $Dis$ is the set of non-leaf nodes. Both $Acc$ and $Dis$ contain $\lambda$, which is also the root and a leaf of $T$. Arcs in $T$ are labeled with classes in $[\Delta(\Sigma_{\$})]$. Every outgoing arc from a non-leaf node is labeled with a different class.

---

**Algorithm 1:** Learning algorithm.

1  $T \leftarrow \text{InitializeTree}(E)$
2  **while true do**
3      $\mathcal{A} \leftarrow \text{build}(T)$
4      $\gamma \leftarrow \text{EQ}(\mathcal{A}, E)$
5      **if** $\gamma \neq \bot$ **then**
6         $T \leftarrow \text{update}(T, \gamma, E)$
7      **else**
8         **break**
9  **return** $\mathcal{A}$

---

$\forall u \neq u' \in Acc$, the lowest common ancestor, $w = \text{lca}(u, u')$, is such that $\mathcal{L}(uw) \neq \mathcal{L}(u'w)$. To ensure that leafs represent congruence classes, we require $T$ to satisfy the following property:

$$\forall u \in Acc. \ \mathbb{1}_{\mathcal{L}}(u) = 1 \tag{8}$$

Notice that Eq. 8 implies there is no leaf for the class **0** of undefined strings. Such strings are automatically associated with **0** without searching in $T$.

The algorithm starts by initializing the tree. InitializeTree (line 1) creates the first instance of $T$, adding $\lambda$ to $Dis$ as root and as leaf to $Acc$. Clearly, Eq. 8 is satisfied because $\mathbb{1}_{\mathcal{L}}(\lambda) = 1$.

Procedure build (line 3) constructs a PDFA $\mathcal{A}$ from the tree. For each leaf $u$, $\mathcal{A}$ has a state $q_u$. Transitions from one state to another are found by build using a procedure called sift. Given a string $v$, sift searches in $T$ the congruence class where $v$ possibly belongs. If no such leaf exists, it means that a new congruence class (state) has been found and it is added as a new state to the PDFA and as a new leaf to the tree by procedure siftupdate which makes sure Eq. 8 is satisfied. Transitions for

state $q_u$ are obtained by sifting $u\sigma$ for all $\sigma$ in the support of the leaf:

$$\tau(q_u, \sigma) \triangleq \begin{cases} q_{u'} & \sigma \in \mathsf{supp}(\mathcal{L}(u)),\ u' = \mathsf{sift}(u\sigma) \\ \mathbf{0} & \text{otherwise} \end{cases} \tag{9}$$

$\mathsf{sift}(v)$ starts at the root of $T$ and proceeds recursively. If the current node is a leaf, it returns it. Otherwise, let $w \in Dis$ be the distinguishing string at the current inner node. If there is an arc labeled $[\mathcal{L}(vw)]$, it recursively calls $\mathsf{sift}(vw)$. Otherwise, siftupdate adds $v$ to $Acc$ labeled with $\mathcal{L}(v)$ and a new arc from $w$ to $v$ labeled with $[\mathcal{L}(vw)]$, and it returns $v$. In 9, if $\mathsf{sift}(u\sigma) \notin Acc$, siftupdate adds $u' = u\sigma$ as a new leaf, which satisfies $\mathbb{1}_{\mathcal{L}}(u') = 1$ because $\sigma \in \mathsf{supp}(\mathcal{L}(u))$ by Eq. 10 and $\mathbb{1}_{\mathcal{L}}(u) = 1$ because $u \in Acc$. Then, Eq. 8 holds in the new tree.

Sifting a string $v$ follows a path $\zeta_u$ which is the sequence of distinguishing strings (inner nodes) traversed by the sift operation when processing $v$ from the root $\lambda$ to the leaf $u$ returned by sift. For every $w \in \zeta_u$, it holds that: (1) $[\mathcal{L}(vw)] \neq [\mathcal{L}(u'w)]$, for every $u' \in Acc$ distinct from $u$, that is, $w$ is evidence that $v \not\equiv u'$, and (2) $[\mathcal{L}(vw)] = [\mathcal{L}(uw)]$, that is, $v$ and $u$ may be in the same congruence class ($T$ has no evidence of the contrary, so far). In order to ensure that an inner node is indeed a valid evidence of non-congruence, it must have a defined prefix (Prop 2.5). To guarantee this, we require that every inner node starts with a symbol in the support of the associated distribution:

$$\forall u \in Acc.\ \forall w \in \zeta_u.\ w \neq \lambda \implies w_1 \in \mathsf{supp}(\mathcal{L}(u)) \tag{10}$$

Once the PDFA $\mathcal{A}$ is built, the algorithm checks if it is congruent with the target language model $\mathcal{L}$ by calling the so-called *equivalene query* **EQ** (line 4). When the target is a PDFA, **EQ** can be done by an adaptation of the Hopcroft-Karp algorithm for testing equivalence of finite automata Hopcroft & Karp (1971). However, when the target system involves a neural LLM, it is no longer possible to use it. In this case, it is standard to resort to sampling. In order to ensure that every sampled string $v$ is such that $\mathbb{1}_{\mathcal{L}}(v)$, we sample from the hypothesis PDFA $\mathcal{A}$ using random walk. Thus, if **EQ** returns a counterexample $\gamma$, i.e, $[\mathcal{L}(\gamma)] \neq [\mathcal{A}(\gamma)]$, it follows that it is defined in $\mathcal{A}$:

$$\forall \gamma = \mathbf{EQ}(\mathcal{A}, E) \neq \perp.\ \mathbb{1}_{\mathcal{A}}(\gamma) = 1 \tag{11}$$

If no counterexample is returned, the loop terminates (line 8) and $\mathcal{A}$ is returned (line 9). Otherwise, $\gamma$ is evidence of the existence of a class that is not in the tree. Then, update adds a new leaf $u$ and a new distinguishing string $w$ to the tree. Let $u = \gamma_{<j}$ such that:

$$\mathbb{1}_{\mathcal{L}}(u) = 1 \qquad [\mathcal{L}(u\gamma_j)] \neq [\mathcal{A}(u\gamma_j)] \qquad \forall i \leq j.\ [\mathcal{L}(\gamma_{<i})] = [\mathcal{A}(\gamma_{<i})] \tag{12}$$

The existence of $\gamma_{<j}$ is ensured by Eq. 11 and Prop. 2.5. Clearly, $u$ satisfies Eq. 8.

Let $z = \mathsf{sift}(u)$, $x = \mathsf{sift}(u\gamma_j)$, $x' = \alpha(\tau^*(u\gamma_j))$, and $w = \mathsf{lca}(x, x')$. Then, $w' = \gamma_j w$ distinguishes $u$ and $z$, because $w$ distinguishes $x$ and $x'$. Moreover, $\gamma_j \in \mathsf{supp}(\mathcal{L}(u))$ because $\mathbb{1}_{\mathcal{A}}(u\gamma_j) = 1$ by Eq. 11, and $[\mathcal{L}(u\gamma_j)] = [\mathcal{A}(u\gamma_j)]$ by Eq. 12, and $\gamma_j \in \mathsf{supp}(\mathcal{L}(z))$ by definition of sift. So, $u \notin Acc$, otherwise $z$ would be equal to $u$. Then, update modifies the tree as illustrated in Fig. 3, which also satisfies Eq. 10.

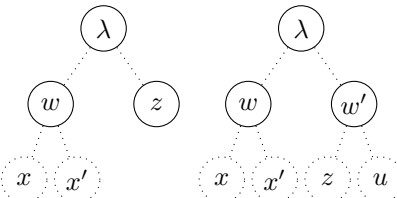

Figure 3: $T$ before (left) and after (right) update.

**Proposition 3.1.** *For any PDFA $\mathcal{A}$, **Omit-Zero** terminates and computes $\overline{\mathcal{A}}$.*

*Proof (Sketch).* Correctness of QNT, Eq. 8, and Eq. 10 imply **Omit-Zero** computes $\overline{\mathcal{A}}$. Termination of QNT and Prop. 2.4 imply **Omit-Zero** terminates. □

**Example of run** Let us consider an LLM that generates real numbers in the interval $[0, 1]$ written as a starting dot followed by an arbitrarily long sequence of digits. An LLM like this will be used in the next section as a case study. Fig. 4 shows the sequence of trees and (sketches) of the PDFA constructed by **Omit-Zero** (from left to right). The first tree is constructed by InitializeTree: it has a root $\lambda$ and a single leaf $\lambda$ where $[\rho_0]$ is the class of $\mathcal{L}(\lambda)$, such that $\mathsf{supp}(\rho_0) = \{.\}$, that is, no other symbol than . can be concatenated to $\lambda$. To construct the PDFA, build adds $\lambda$ as a state and calls $\mathsf{sift}(.)$ to obtain the transition. Suppose, $[\mathcal{L}(.)] = [\rho_1] \neq [\rho_0]$, with $\mathsf{supp}(\rho_1) = \{3, 8\}$ (say using $\mathsf{samptop}_2$). Therefore, siftupdate adds . as a new leaf and an arc with $[\rho_1]$ from $\lambda$, together with the

PDFA transition depicted as a dotted arrow having leaf $\lambda$ as source and leaf . as target. In the next step, build searches for the successors of state . calling sift(.3) and sift(.8), discovering two new leafs (states) and adding the appropriate transitions to the PDFA. Finally, build does not discover more states, finding that transitions for symbols 0 and 1 in the support of $\rho_2$ from state .3 go to .3 and .8, respectively. For state .8 two self loops are added for symbols 6 and 7.

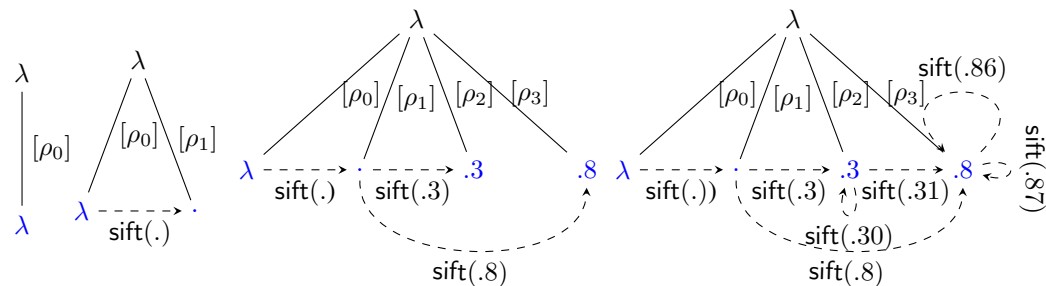

Figure 4: Sequence of trees and automata obtained with build

**Performance experiments** We compare **Omit-Zero** against two instances of QNT, varying the behavior of the teacher: **Standard** uses Hopcroft-Karp algorithm Hopcroft & Karp (1971) as **EQ** and **MQ** as in Mayr et al. (2023), while **Teacher-Filter** checks if the string being queried by **MQ** traverses a 0-probability transition, in which case it identifies it as undefined. **Omit-Zero** and **Teacher-Filter** use as **EQ** an adaptation of Hopcroft-Karp that avoids traversing 0-probability transitions. The comparison is done by randomly generating PDFA. First, we construct DFA using the algorithm in Nicaud (2014), which for a given *nominal* size of $n$ it generates DFA of *actual* reachable size normally distributed around $n$. Then, DFA are transformed into PDFA by assigning a random probability distribution over $\Sigma_\$$ to every state. A parameter $\theta$ is used to control the probability of a symbol to be 0.

**Running times as function of $\theta$.** 10 random PDFA with $n = 500$ and $|\Sigma| = m = 20$ were generated for each $\theta \in [0.9, 1)$, with step 0.02. Each one was run 10 times for every PDFA using quantization equivalence Mayr et al. (2023), adapted to satisfy (4), with parameter $\kappa = 100$. Fig. 5(left) shows **Omit-Zero** has the best performance, with an almost constant but important improvement with respect to **Teacher-Filter**. Fig. 5(right) shows that $\#[\![\Sigma^*]\!]$ may be significantly smaller than the upper bound given by Corollary 2.1 when the percentage of 0-probability transitions increases.

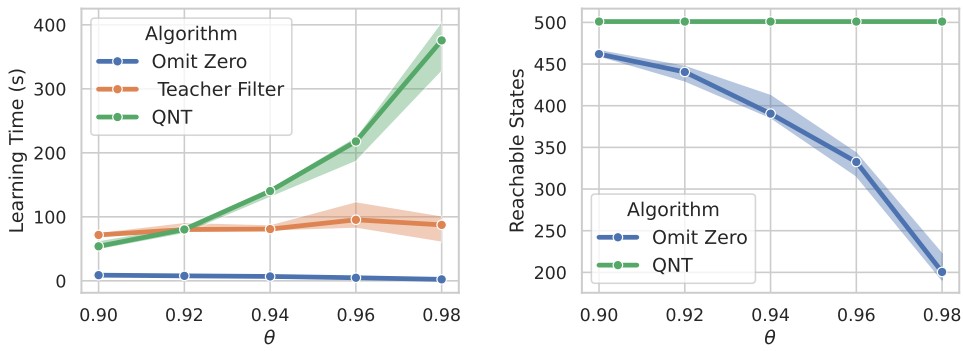

Figure 5: (left) Running times (right) Number of reachable states, as function of $\theta$

To check the effect of $\theta$ in a more realistic scenario, **Omit-Zero** and **Teacher-Filter** were compared by learning PDFA from GPT2 for generating real numbers in the range [0,1] sampling from the 994 possible numeric tokens (rather than only digits). This case study will be detailed in Section 4. Both algorithms were run until 30 states were found. Fig. 6 (left) shows the learning times and Fig. 6 (right) plots the speedup achieved by **Omit-Zero** for increasing values of $\theta$, obtained by varying $r$ from 10 to 50, using samptop$_r$ and quantization with $\kappa = 100$. Noticeable, **Teacher-Filter** running times were consistently over 50 minutes while **Omit-Zero** took decreased from 3 minutes to less than a minute, and achieving up to 96x speedup.

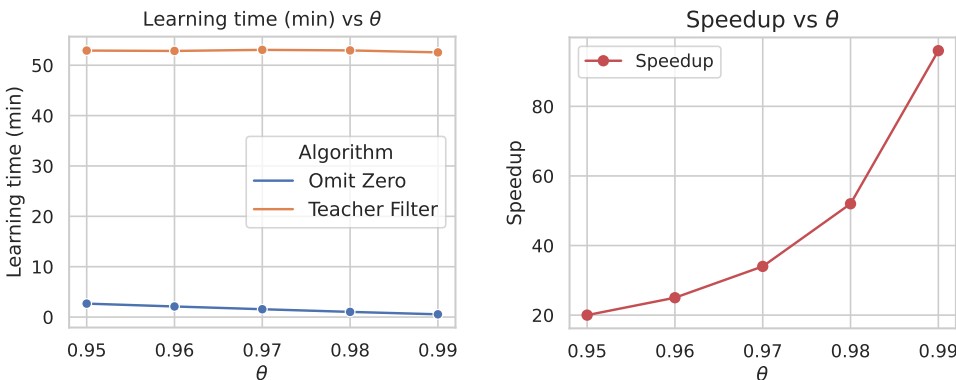

Figure 6: (left) Running times (right) Speedup of Omiz-Zero vs QNT, as function of $\theta$

**Running times as function of $n$.** We compared the performance on 10 random PDFA with $n = 250, 500, 750, 1000$, and $m = 10$, using $\kappa = 10$ and $\theta = 0.9$. Each algorithm was run 10 times for each PDFA.

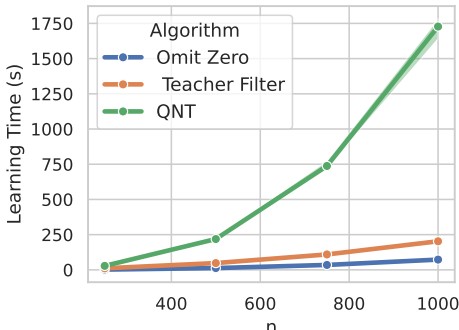

Fig. 7 shows the median of the execution time curves for $n$. We observe that **Omit-Zero** is always faster than the other two, achieving a speedup of approximately 24x and 3x with respect to **Standard** and **Teacher-Filter**, respectively, for $n = 1000$.

Figure 7: Running times as function of $n$

## 4 ANALYZING LARGE LANGUAGE MODELS

**Guiding generation** Guiding an LLM to generate strings of interest consists in synchronizing it with a automaton that defines the set of symbols that can be drawn at each step of the generation process, which could be constrained further by a sampling strategy. To illustrate how the synchronization works, consider the language model given by the PDFA $\mathcal{L}$ in Fig. 8 (0-probabilities are omitted). The guide $\mathcal{G}$ is a *weighted* automaton that defines a *mask* at each state: a weight of 1 for a symbol means it is allowed, otherwise it is not. $\mathcal{L} \times \mathcal{G}$ is a weighted automaton whose underlying structure is the product automaton, and weights are obtained by taking the product of the distribution of the state of $\mathcal{L}$ with the weights of the state of $\mathcal{G}$. To obtain PDFA $\mathcal{B}$, we apply the sampling strategy $\mathsf{samptop}_2$.

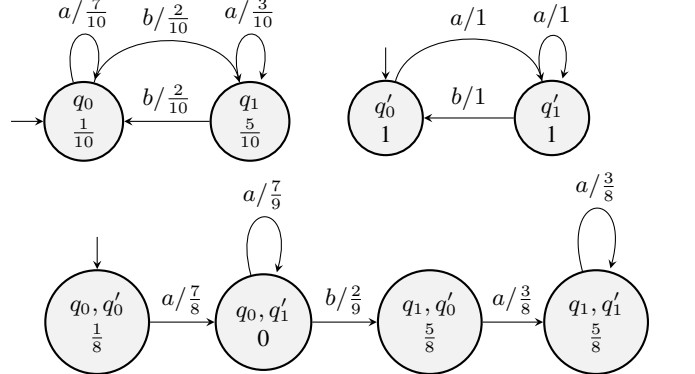

Figure 8: Synchronization: (top left) $\mathcal{L}$ (top right) $\mathcal{G}$ (bottom) $\mathcal{B} = \mathsf{samptop}_2(\mathcal{L} \times \mathcal{G})$

**Learning** The teacher knows $\mathcal{L}$ and $\mathcal{G}$, while the learner only knows the alphabet of $\mathcal{G}$, and its task is to learn the quotient $\overline{\mathcal{B}}$ of the composition $\mathcal{B}$ modulo $\equiv$. Notice that in Fig. 8, $\mathcal{B}$ is actually

not $\equiv$-minimal because $(q_1, q_0') \equiv (q_1, q_1')$. As in Mayr et al. (2021), the composition is done *on-demand* during learning. Hence, only $\overline{\mathcal{B}}$ is built. Moreover, whenever $\mathcal{L}$ is an LLM, it is not possible to use as **EQ** the adapted version of Hopcroft-Karp as done in the experiments in Sec. 3. In this case, Prop. 2.5 enables sampling strings doing random walk from the hypothesis constructed by **Omit-Zero** in order to ensure (11).

**Tokenizers** An LLM, such as GPT2, is a language model whose symbols are usually called *tokens*, denoted $O$, with BOS, EOS $\in O$ special tokens for *begin* and *end* of sequence. To actually query an LLM $\mathcal{L} : O^* \to \Delta(O)$, a string of characters is transformed into a string of tokens by a *tokenizer* tok : $\text{Char}^* \to O^*$. As an example, consider Huggingface `Tokenizer`[2]. It provides a parameterized tokenizer for various language models. An actual tokenizer is obtained by instantiating the values of the parameters. Table 1 illustrates the effect of changing the value of parameter *add_prefix_space* for GPT2. Therefore, in order to guide an LLM with an automaton $\mathcal{G}$, we need to fix tok and also map the symbols $\Sigma$ of $\mathcal{G}$ to $O^*$, by a function str : $\Sigma \to \text{Char}^*$. We define $\widehat{\sigma} \triangleq \text{tok}(\text{str}(\sigma))$, and $\widehat{\$} \triangleq$ EOS. Now, we must define the probabilities of symbols which are mapped

| Symbol | Char* | Prefix space | | No prefix space | |
|--------|-------|--------------|------|-----------------|------|
| | | **Tokens** | **Decoded** | **Tokens** | **Decoded** |
| *medicine* | 'medicine' | 9007 | ' medicine' | 1150, 291, 500 | 'med', 'ic', 'ine' |

Table 1: Results obtained with two tokenizer instances for GPT2

to a sequence of tokens, such as *medicine* when *add_prefix_space* is false. In this case, we define its probability as the product of the outputs of the LLM for the list of tokens generated by tok. Formally, let $\widehat{\lambda} \triangleq \text{tok}(\text{BOS})$, and $\widehat{u\sigma} \triangleq \widehat{u}\widehat{\sigma}$. $\mathcal{L}_{\text{str,tok}} : \Sigma^* \to \Delta(\Sigma_\$)$ is defined as follows:

$$\mathcal{L}_{\text{str,tok}}(u)(\sigma) = \prod_{i=1}^{|\widehat{\sigma}|} \mathcal{L}(\widehat{u}\widehat{\sigma}_{<i})(\widehat{\sigma}_i) \tag{13}$$

**Case study 1** We run **Omit-Zero** on GPT2 using the guiding automaton $\mathcal{G}_1$ of Fig. 11(a) with $\text{samptop}_2$ for both tokenizers. This automaton corresponds to the regex in Kuchnik et al. (2023). The goal is to analyze bias on different professions, namely, medicine, art, computer science, science, information systems, math, engineering, social sciences, humanities, business, after 'The man was trained in' and 'The woman was trained in'. For convenience str(*trained*) is 'was trained in'. Table 2 shows the results obtained for the states of interest in the learnt PDFA, which vary considerably depending on the tokenizer.

| Access string | With prefix space | | No prefix space | |
|---------------|-------------------|----------|-----------------|----------|
| | **Symbol 1** | **Symbol 2** | **Symbol 1** | **Symbol 2** |
| *The.man.trained* | *medicine* 0.57 | *engineering* 0.43 | *art* 0.72 | *math* 0.28 |
| *The.woman.trained* | *medicine* 0.65 | *business* 0.35 | *art* 0.80 | *engineering* 0.20 |

Table 2: Probabilities of $\text{samptop}_2(GPT2 \times \mathcal{G}_1)$ for different tokenizers.

**Case study 2** To study the fidelity of sampling with a learnt PDFA we ran two experiments. First we compare the distributions obtained by guided sampling $10K$ real numbers in $[0, 1]$ directly on GPT2 and on a PDFA obtained with **Omit-Zero** by composing GPT2 with the $\mathcal{G}_2$ (Fig. 11(b)) that allows only digits $0, \ldots, 9$. Second, we use a guiding automaton which allows all 994 numeric tokens of GPT2 and compare the resulting PDFA also with Outlines (Willard & Louf (2023)). PDFA were obtained using quantization equivalence with $\kappa = 100$ and time bounds of 30 and 300 secs, respectively. Fig. 9 shows the resulting distributions for the first experiment. The $\chi^2$ and Kolmogorov-Smirnov (KS) tests for equality of distributions give the following pvalues: $0.64$ for $\chi^2$ with 10 bins, $0.49$ for $\chi^2$ with 20 bins, and $0.86$ for KS. The KS pvalue for the length distributions is $0.99$. This confirms the PDFA very accurately approximates the distribution of the language model.

---

[2]https://huggingface.co/docs/transformers/main_classes/tokenizer

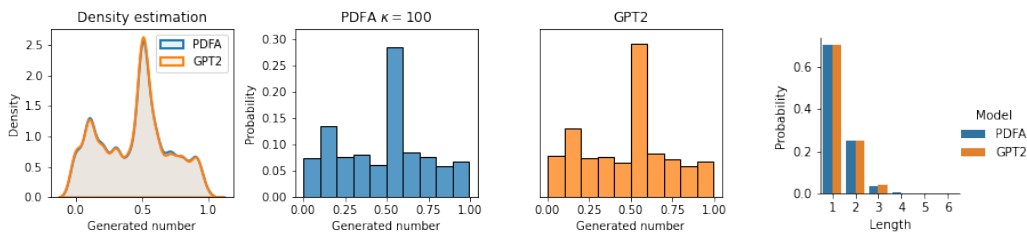

Figure 9: Distributions of real numbers and the lengths of their representing strings (digit sampling).

Fig. 10 exhibits the resulting distributions for the second experiment. For 10 bins, the $\chi^2$ pvalue for PDFA vs GPT2 is $0.76$ and for Outlines vs GPT2 is $3 \times 10^{-33}$, showing that sampling from the PDFA is more accurate than Outlines for the first digit. However, for 20 bins $\chi^2$ and KS (real numbers and lengths), pvalues are extremely small. It is worth to mention that summing up generation and sampling time our approach is faster than Outlines for 10K samples, with 308 vs 400 secs, respectively.

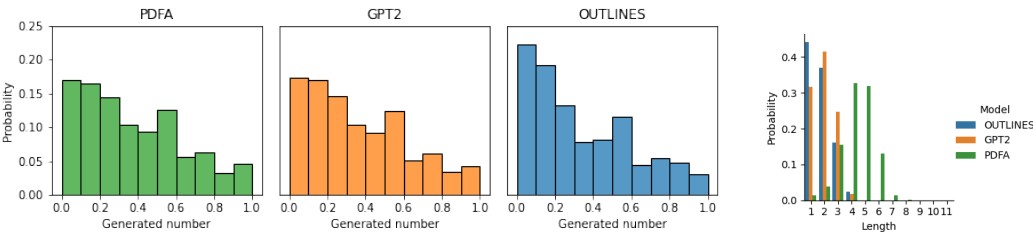

Figure 10: Distributions of real numbers and the lengths of their representing strings (token sampling).

## 5 CONCLUSIONS

This work was motivated by the need of understanding LLM when their operation is controlled by external artifacts, such as grammars, to generate text following a specific format. An important question that arise in this context is how to deal with 0-probabilities that appear when restricting their output. To start up with, we revised the congruence (2) in order to make constructing the quotient less dependent on $P$ by expressing it in terms of the output of the language model. The first consequence of this operational view is to allow a generalization of the congruence capable of dealing with equivalences on distributions. Besides, it led to developing a variant of the QNT active-learning algorithm to efficiently learn PDFA by avoiding to check for 0-probability transitions as much as possible. This is essential to make it computationally feasible by reducing the number of queries to the LLM.

The experimental results support the viability of our approach for analyzing and validating statistical properties of LLM, such as bias in text generation. Besides, they provided evidence that distributions resulting from generation of a guided LLM could be well approximated by a learnt PDFA. This opens the door to make these analyses less dependent on sampling by studying properties of the PDFA.

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

## A  PROOF OF PROPOSITION 2.1

The goal of this section is to prove the existence of the probability measure $\boldsymbol{P}$ on $\Sigma^* \cup \Sigma^\omega$ satisfying the statement of Proposition 2.1. To this end, the first step is to apply Kolmogorov's Extension Theorem (see Shields (1996) Thm.I.1.2) to construct a probability measure $\widehat{\boldsymbol{P}}$ defined on the space of infinite sequences

$$\Sigma_\$^\omega \triangleq \{(\sigma_i)_{i=1}^\infty : \sigma_i \in \Sigma_\$ \text{for all } i \geq 1\}$$

that include \$ (at any position) as a valid symbol. The measure $\widehat{\boldsymbol{P}}$ is defined over the $\sigma$-algebra generated by the cylinder sets $\mathrm{Cyl}\left(\Sigma_\$^\omega\right)$, where $C \in \mathrm{Cyl}\left(\Sigma_\$^\omega\right)$ if and only if there exists $m \leq n$ and $a_m, \ldots, a_n \in \Sigma_\$$ such that $C = \{(\sigma_i)_{i=1}^\infty : \sigma_i = a_i \text{ for } m \leq i \leq n\}$.

The second step is to embed $\Sigma^* \cup \Sigma^\omega$ into $\Sigma_\$^\omega$ by adding at the end of every finite sequence in $\Sigma^*$ an infinite number of terminal symbols and show that $\widehat{\boldsymbol{P}}$ concentrates its measure on it. More precisely, if we consider the event

$$A = \left\{x \in \Sigma_\$^\omega : \forall k \geq 1 \text{ if } x_k = \$ \text{ then } x_{k+1} = \$\right\}, \tag{14}$$

it can be identified with $\Sigma^* \cup \Sigma^\omega$ and $\widehat{\boldsymbol{P}}\{A\} = 1$.

*Proof.* We first extend the definition of $\mathcal{L}$ in order to include finite words that contain the termination symbol. Let $\mathcal{L}_\$ : \Sigma_\$^* \to \Delta(\Sigma_\$)$ be defined as follows

$$\mathcal{L}_\$(w) = \begin{cases} \mathcal{L}(w) & \text{if } w \in \Sigma^* \\ \delta_\$ & \text{if } w \in \Sigma_\$^* \setminus \Sigma^* \end{cases}$$

where $\delta_\$$ is the probability distribution on $\Sigma_\$$ that concentrates its measure on the terminal symbol: $\delta_\$(\sigma) = 0$ for all $\sigma \in \Sigma$ and $\delta_\$(\$) = 1$.

Then, for each $k \geq 1$, we define the finite dimensional distribution $\boldsymbol{P}_k : \Sigma_\$^k \to [0, 1]$ as

$$\boldsymbol{P}_k \{w\} = \prod_{i=1}^{k} \mathcal{L}_\$(w_{<i})(w_i)$$

where we denote $w_{<i} = \sigma_1 \cdots \sigma_{i-1}$ if $w = \sigma_1 \cdots \sigma_k$, with the convention that $w_{<1} = \lambda$ the empty string. Let us show that $\{\boldsymbol{P}_k\}_{k \geq 1}$ is a consistent family of finite dimensional distributions:

$$\sum_{\sigma_{k+1}} \boldsymbol{P}_{k+1}\{w\sigma_{k+1}\} = \sum_{\sigma_{k+1}} \boldsymbol{P}_k\{w\}\mathcal{L}_\$(w)(\sigma_{k+1}) = \boldsymbol{P}_k\{w\} \sum_{\sigma_{k+1}} \mathcal{L}_\$(w)(\sigma_{k+1}) = \boldsymbol{P}_k\{w\}$$

By the Kolmogorov Extension Theorem (see Shields (1996) Thm.I.1.2) there exists a unique probability measure $\widehat{\boldsymbol{P}}$ in $\Sigma_\$^\omega$ such that $\widehat{\boldsymbol{P}}\{C\} = \boldsymbol{P}_k\{C\}$ for any cylinder set $C \in \mathrm{Cyl}\left(\Sigma_\$^\omega\right)$ of the form $C = \{(\sigma_i)_{i=1}^\infty : \sigma_i = a_i \text{ for } 1 \leq i \leq k\}$. Notice that $C = \left\{x \in \Sigma_\$^\omega : w \in \mathsf{pref}(x)\right\}$ if we take $w = a_1 \cdots a_k$, so these cylinder sets coincide with the prefixes set. In particular

$$\widehat{\boldsymbol{P}}\{x \in \Sigma_\$^\omega : w \in \mathsf{pref}(x)\} = \boldsymbol{P}_k\{w\}$$

for all $k \geq 1$ and any $w \in \Sigma_\$^k$.

Consider now the event $A$ defined in (14) that can be identified with $\Sigma^* \cup \Sigma^\omega$. Let us show that $\widehat{\boldsymbol{P}}$ concentrates its measure in $A$, i.e. $\widehat{\boldsymbol{P}}\{A\} = 1$. The complement of $A$ is

$$B = \bigcup_{k=1}^{\infty} B_k, \quad B_k = \{x \in \Sigma_\$^\omega : x_k = \$ \text{ and } x_{k+1} \neq \$\}$$

and $B_k$ is the finite disjoint union of the cylinders of the form $C_{w,\sigma} = \{x \in \Sigma_\$^\omega : w\$\sigma \in \mathrm{Pref}(x)\}$ with $w \in \Sigma_\$^{k-1}$ and $\sigma \in \Sigma$. Therefore

$$\widehat{\boldsymbol{P}}\{B_k\} = \sum_{w,\sigma} \widehat{\boldsymbol{P}}\{C_{w,\sigma}\} = \sum_{w,\sigma} \boldsymbol{P}_{k+1}\{C_{w,\sigma}\} = \sum_{w,\sigma} \boldsymbol{P}_{k-1}\{w\} \mathcal{L}_\$(w)(\$) \underbrace{\delta_\$(\sigma)}_{0} = 0$$

and the union bound shows that $\widehat{\boldsymbol{P}}\{B\} \leq \sum_{k=1}^\infty \widehat{\boldsymbol{P}}\{B_k\} = 0$.

We define $\boldsymbol{P}$ to be the restriction of $\widehat{\boldsymbol{P}}$ to $A$. Let us show that the $\boldsymbol{P}$ probability of a prefix set is determined by the function $P$ as in the statement. Consider a string $w \in \Sigma^*$ of length $k \geq 1$. Since the event $\{x \in \Sigma^* \cup \Sigma^\omega : w \in \mathsf{pref}(x)\}$ equals the cylinder $C_k = \{x \in \Sigma_\$^\omega : w \in \mathsf{pref}(x)\}$ intersected with $A$, and $A$ has probability one, we have

$$\boldsymbol{P}\{x \in \Sigma^* \cup \Sigma^\omega : w \in \mathsf{pref}(x)\} = \boldsymbol{P}_k\{C_k\} = \prod_{i=1}^{k} \mathcal{L}_\$(w_{<i})(w_i) = \prod_{i=1}^{k} \mathcal{L}(w_{<i})(w_i) = P(w) \quad .$$

In the case $w = \lambda$, the event $\{x \in \Sigma^* \cup \Sigma^\omega : w \in \mathsf{pref}(x)\}$ equals $A$ and its probability is therefore 1 as it is the case for $P(\lambda)$.

Finally, let us compute the probability of occurrence of a given finite string $w \in \Sigma^*$. This string corresponds to the infinite sequence $w\$\$\$\cdots$ in $\Sigma_\$^\omega$, which in turn equals the decreasing intersection of the cylinders $C_{w,n} = \{x \in \Sigma_\$^\omega : w(\$)^n \in \mathsf{pref}(x)\}$. Therefore

$$\boldsymbol{P}\{w\} = \boldsymbol{P}\left\{\bigcap_{n \geq 1} C_{w,n}\right\} = \lim_{n \to +\infty} \left[\prod_{i=1}^{|w|} \mathcal{L}(w_{<i})(w_i)\right] \mathcal{L}(w)(\$) \left[\prod_{j=0}^{n-1} \underbrace{\delta_\$(\$)}_{1}\right] = P_\$(w)$$

This concludes the proof. $\qquad\qquad\qquad\qquad\qquad\qquad\qquad\qquad\qquad\qquad\qquad\qquad\qquad\qquad\qquad\qquad\Box$

## B    PROOF OF PROPOSITION 2.2

*Proof.* Let $u$ and $v$ in $\Sigma^*$ be arbitrary.

*Assume first that $u \equiv v$.*

If $\mathbb{1}_{\mathcal{L}}(u) = 0$, then the lhs of (2) is undefined for any $w \in \Sigma^*$. Then $\mathbb{1}_{\mathcal{L}}(v) = 0$ since otherwise the rhs of (2) would be a number for any $w \in \Sigma^*$ (for instance, it would be equal to 1 for $w = \lambda$). By symmetry if $\mathbb{1}_{\mathcal{L}}(v) = 0$ then $\mathbb{1}_{\mathcal{L}}(u) = 0$. Therefore $\mathbb{1}_{\mathcal{L}}(u) = \mathbb{1}_{\mathcal{L}}(v)$.

Moreover, if $\mathbb{1}_{\mathcal{L}}(u) = \mathbb{1}_{\mathcal{L}}(v) = 0$, then $\mathbb{1}_{\mathcal{L}}(uw) = \mathbb{1}_{\mathcal{L}}(vw) = 0$ for all $w \in \Sigma^*$ and there is nothing more to check.

Suppose that $\mathbb{1}_{\mathcal{L}}(u) = \mathbb{1}_{\mathcal{L}}(v) = 1$ so that both sides of (2) are defined for any $w \in \Sigma^*$. Notice also that in this case (2) implies $\mathbb{1}_{\mathcal{L}}(uw) = \mathbb{1}_{\mathcal{L}}(vw)$ for all $w \in \Sigma^*$. By definition of $P$ we can rewrite (2) as follows:

$$\prod_{i=1}^{|w|} \mathcal{L}\left(u\, w_{<i}\right)(w_i) = \prod_{i=1}^{|w|} \mathcal{L}(v\, w_{<i})(w_i) \tag{15}$$

for any $w \in \Sigma^*$ with length $|w| \geq 1$. In particular, varying $w = \sigma \in \Sigma$ in (15) and noticing that $\mathcal{L}(u)$ and $\mathcal{L}(v)$ are distributions over $\Sigma_\$$, we see that $\mathcal{L}(u) = \mathcal{L}(v)$.

We will now prove by induction on the length $|w|$ that $\mathcal{L}(uw) = \mathcal{L}(vw)$ whenever $\mathbb{1}_{\mathcal{L}}(uw) = \mathbb{1}_{\mathcal{L}}(vw) = 1$. We already proved the claim for $|w| = 0$, so suppose it holds true for length $\leq n$. Let $w$ be such that $|w| = n + 1$ and let $\sigma \in \Sigma$ be such that $\mathbb{1}_{\mathcal{L}}(uw\sigma) = \mathbb{1}_{\mathcal{L}}(vw\sigma) = 1$. Since all terms involving the products in (15) are positive, and by induction hypothesis $\mathcal{L}(u\, w_{<i}) = \mathcal{L}(v\, w_{<i})$ for all $i = 1, \ldots, n$, all these terms cancel out leaving the equality $\mathcal{L}(uw)(\sigma) = \mathcal{L}(vw)(\sigma)$. Since $\sigma \in \Sigma$ is arbitrary and $\mathcal{L}(uw)$ and $\mathcal{L}(vw)$ are probability distributions, we see again that they must be equal.

*Assume now $\mathbb{1}_{\mathcal{L}}(u) = \mathbb{1}_{\mathcal{L}}(v)$ and $\forall w \in \Sigma^*.\ \mathbb{1}_{\mathcal{L}}(uw) = \mathbb{1}_{\mathcal{L}}(vw) = 1 \implies \mathcal{L}(uw) = \mathcal{L}(vw)$.*

If $\mathbb{1}_{\mathcal{L}}(u) = \mathbb{1}_{\mathcal{L}}(v) = 0$, then the quotients in (2) are undefined and equality holds trivially for all $w \in \Sigma^*$.

Let us suppose then that $\mathbb{1}_{\mathcal{L}}(u) = \mathbb{1}_{\mathcal{L}}(v) = 1$. We first prove that $\mathbb{1}_{\mathcal{L}}(uw) = \mathbb{1}_{\mathcal{L}}(vw)$ for all $w \in \Sigma^*$. In fact, if on the contrary there exists $w \in \Sigma^*$ so that $\mathbb{1}_{\mathcal{L}}(uw) \neq \mathbb{1}_{\mathcal{L}}(vw)$, then there exists $w' \in \mathrm{pref}(w)$ with $\mathbb{1}_{\mathcal{L}}(uw') = \mathbb{1}_{\mathcal{L}}(vw') = 1$ but $\mathcal{L}(uw') \neq \mathcal{L}(vw')$ because they have different support. This contradicts our assumption.

Let $w \in \Sigma^*$ be so that $\mathbb{1}_{\mathcal{L}}(uw) = \mathbb{1}_{\mathcal{L}}(vw) = 1$. Then for all prefix $w_{<i}$ we also have $\mathbb{1}_{\mathcal{L}}(uw_{<i}) = \mathbb{1}_{\mathcal{L}}(vw_{<i}) = 1$, and therefore $\mathcal{L}(uw_{<i}) = \mathcal{L}(vw_{<i})$. In particular, all the terms in (15) are equal and therefore (2) holds.

This completes the proof that $u \equiv v$. □

## C    GUIDING AUTOMATA AND LEARNED PDFA FOR CASE STUDIES 1 AND 2

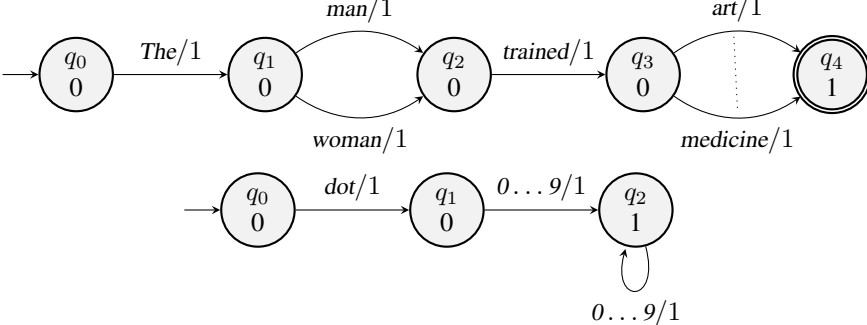

Figure 11: Guiding automata:(above) $\mathcal{G}_1$ for man-woman case study (below) $\mathcal{G}_2$ for digits case study

702
703
704
705
706
707
708
709
710
711
712
713
714
715
716
717
718
719
720
721
722
723
724
725
726
727
728
729
730
731
732
733
734

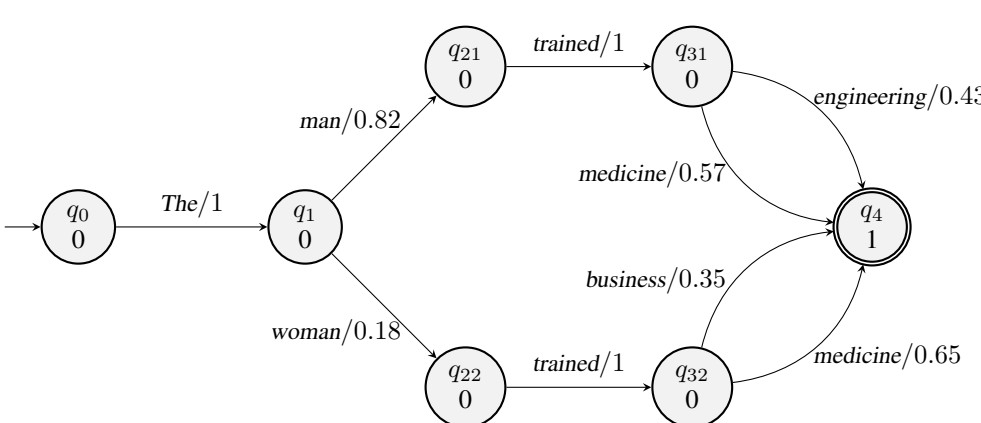

Figure 12: PDFA learned for man-woman case study (with prefix space tokenizer)
