# OpenReview forum: "A probablistic automata learning approach for analyzing and sampling constrained LLM"
_ICLR.cc/2025/Conference — Submitted to ICLR 2025_

### Official Review · Reviewer_hTvK · 2024-10-31

**Soundness:** 3
**Presentation:** 3
**Contribution:** 3
**Rating:** 6
**Confidence:** 3

**Summary:**

Many automata theoretic techniques have been introduced to analyse neural sequence-processing recognizers  by composing them with automata theoretic formalisms with the purpose of verifying properties on-the-fly. In this work the authors propose an approach to combine automata theoretic formalisms with language generators, such as neural language models, in order to guide the generation process or constrain the text generation with some common sampling strategies. In this context, the occurrence of symbols that have zero probability of appearing is a problem. For example, the generation may not terminate, or the model may not define a probability distribution over finite strings.

In this paper, the authors define a notion of  Myhill-Nerode-like congruence over strings which takes into account the occurrence of zero-probabilities, that provides an underlying formal basis for learning of probabilistic deterministic finite automata (PDFA) from neural language models constrained both by by automata and sampling strategies. Another contribution is that they propose a new algorithm for learning the quotient with respect to this congruence. The authors provide experimental evidence that the new approach has some advantages with respect to existing approaches. Finally, the authors providea framework to analyze statistical properties of LLMs.

**Strengths:**

In my opinion, the paper is quite interesting. There is currently an impending need to understand the behavior of LLMs when their operation is controlled by external such as automata and more general formalisms. In this work the authors provide a contribution in this direction.

**Weaknesses:**

The paper as a whole is a whole is a fair contribution of the topic of LLMs. Nevertheless, the results follow by extending and adapting known constructions within automata theory. In this sense, I did not find the results of the paper particularly surprising.

**Questions:**

I do not have questions.

---

### Official Review · Reviewer_sah2 · 2024-11-04

**Soundness:** 3
**Presentation:** 2
**Contribution:** 3
**Rating:** 5
**Confidence:** 3

**Summary:**

In this paper, the authors consider the problem of learning a probabilistic deterministic finite automaton (PDFA) by interacting with a constrained sequence model with a special sampling scheme, such as a large language model constrained with a given property and sampled with a top-k selection scheme. The main challenge to learn a PDFA in such a setting is that the distribution over strings by the constrained LLM with a particular sampling scheme may assign non-zero probabilities to infinite sequences; in order to even describe an ideal PDFA to learn, one has to come up with an appropriate notion of equivalence on strings that are induced by the constraint and the sampling scheme. In the paper, the authors indeed propose such a notion in a clean general way, define the induced quotienting operators on language models (formalised as maps from finite strings to probability distributions over characters and EOS), and PDFAs, and analyse the properties of these operators. Based on this formal development, they propose an algorithm for learning a PDFA from interactions with a constrained language model with a special sampling scheme formalised by an equivalence relation on distributions over characters and EOS. Their algorithm is applied to simple language models which are based on randomly generated DFAs, and also to the constrained GPT-2 model with a special sampling scheme. The results of these experiments show the promise of the algorithm in the paper.

**Strengths:**

1. This paper describes the issue of a constrained language model assigning non-zero probability to infinite sequences. My understanding is that this is not one of the commonly discussed issues on language models (and their distillations which seem to be related to the work in the paper). Asking an unusual question, I think, is good for the research area.

2. The formal development in the paper is thorough and rigorous. Here I mean Section 2 of the paper, which I liked and learnt a lot from. But as I will mention in the weakness box below, I found Section 3 of the paper confusing and hard to follow.

3. The authors' learning algorithm is derived from a solid theory.

**Weaknesses:**

1. My main reservation is that Section 3 was very difficult to follow. It is the part that describes the main contribution of the paper, namely, the algorithm for learning a PDFA from the interactions with a teacher model (formalised via the EQ query). But the section assumes the familiarity with QNT, and does not describe what sift, build, EQ, update, and InitializeHypothesis do, and how their learning algorithm works . Also, I couldn't understand what path means (which is used in (8)). Illustrating the run of the algorithm with a concrete example may be helpful.

2. The next point is not really a weakness, but it partly explains why I am not a strong supporter of the paper. While I read the paper, one question kept popping up to me: what can one do if she or he has a quotiented PDFA that models a given constrained language model with a special sampling scheme (such as top-k)? The abstract says that such a quotiented PDFA can be used to analyse the statistical properties of the language model. Elaborating on this and giving some concrete examples would make the paper more appealing to someone like me.

**Questions:**

If the authors can respond to the two points that I mentioned in the weakness box, that would be helpful for me to understand the paper more. But they don't have to do so.

---

> ### Author Response · Authors · 2024-11-23
> **Response to weakness 1**
>
> We added a global response to this comment https://openreview.net/forum?id=CIcMuee69B&noteId=KoKOiS3cK8
>
> We hope it adequately addresses your concerns.

---

> ### Author Response · Authors · 2024-11-23
> **Response to weakness 2**
>
> Actually, the case studies are intended to give concrete examples of the use of the learned PDFA. We will try to make their purpose more clear.
>
> It is important to remark that the behavior of an application that embeds an LLM does not depend solely on the LLM, but also on the software harness around it, which includes the tokenizer that transforms human language into tokens, as well as the sampling strategy. The purpose of Sections 2 and 3 is to propose a formal way to deal with 0 probabilities inevitably introduced when constraining the output of the LLM. The goal of Section 4 is to illustrate its application in realistic settings (as studied in other works of the literature such as [5, 14]), for which it is necessary to find a formal way to take into account the tokenizer. Our approach allows considering a sentence in human language as a symbol and relate it to the sequence of tokens depending on the tokenizer, which enables the verifier to define the appropriate level of abstraction and to seamlessly compare different configurations (including different LLM, tokenizers and sampling strategies).
>
> In the first case study, the primary goal is to analyze gender bias in GPT2 as proposed in [5], by looking at the probability distributions of several professions after the words “man” and “woman”. The learned PDFA exhibits the probability distribution of the considered professions for each gender. It shows that for both configurations the probability distributions have bias in gender, and that this bias is different for each configuration. In contrast to other works in the literature this is done without resorting to sampling or evaluating on a dataset.
>
> Now, from the perspective of model-based analysis, the LLM together with the guide, the sampling strategy, and all of the configuration elements, such as the tokenizer, is the system under analysis, and the learned PDFA is the model of such system on which the analysis will be carried on. In this case, we need to ensure that the learned model is faithful, in the sense that what is concluded on the model can be extrapolated to the system under analysis.
>
> To assess this, the goal of the second case study is to evaluate how close the distribution represented by the learned PDFA is to the one defined by the target system, where GPT2 is guided to generate real numbers in the interval [0…1] as proposed in [14]. Besides, we illustrated how the task could actually be performed using different ways of guiding the LLM. First, with a guide that only allows the LLM to produce one digit at a time, and second, enabling it to pick any of its numeric tokens. Since the PDFA were learned using an equivalence relation between distributions, in this case quantization, it is natural to study to what extent this abstraction leads to a quantifiable divergence with respect to the distribution of the real system. To evaluate this, we perform statistical tests on sets of samples generated by the learned PDFA and the system. In both cases, it is observed that the distributions over the set of numbers of finite length in the interval [0..1] defined by the PDFA approximate quite accurately that of the target system. On the other hand, the experiments revealed that this is also the case for the distribution of lengths when the LLM is guided to generate sequences of digits but not for sequences of numeric tokens.

---

> > ### Comment · Reviewer_sah2 · 2024-11-26
> > **Thanks!**
> >
> > Thank you for your response! It helps me to understand the paper better. I am still reluctant to change my score. If I happen to review the next version of the paper with detailed explanation on various operators, I will give a better score to this paper.

---

> > > ### Author Response · Authors · 2024-11-28
> > > **Section 3**
> > >
> > > We have rewritten Section 3 in the revised version, adding a more detailed explanation of the algorithm and its inner operations, and an example of run inspired by the case study of the generation of real numbers consisting on sequences of digits.

---

### Official Review · Reviewer_gJvq · 2024-11-04

**Soundness:** 3
**Presentation:** 2
**Contribution:** 3
**Rating:** 5
**Confidence:** 4

**Summary:**

The paper introduces a new congruence on words w.r.t. a given a language model $\mathcal{L}:\Sigma^*\to\Delta(\Sigma)$, which can be used to learn surrogate probabilistic deterministic finite automata using a Myhill-Nerode type theorem. It builds heavily on ideas introduced in [6], where a similar congruence and a corresponding tree-based learning algorithm have been developed, but addresses the problem of null probabilities, which can occur if the model output is externally constrained, e.g. by an automaton.

More precisely, the congruence is defined relative to an equivalence relation $E$ on the probability simplex $\Delta(\Sigma)$
$$
u\equiv_E v :\Leftrightarrow \mathbb{1}(u) = \mathbb{1}(v) \land \forall w\in\Sigma^*:  \mathbb{1}(uw) = \mathbb{1}(vw) =1 \to \mathcal{L}(uw) =_E \mathcal{L}(vw),
$$
where $\mathbb{1}(u)$ indicates whether the string $u$ has positive probability under the model.
As mentioned earlier, the main difference to the work done in [6] is the explicit handling of null probabilities. The definition ensures that $0$ probability transitions do not need to be explored while still maintaining a meaningful congruence. As a consequence, the QNT algorithm from [6] can be adopted to avoid $0$-probability transitions as much as possible. This is useful to keep the number of queries to the LLM as low as possible during learning.

The developed algorithm is compared against to other variants of QNT and shows significantly reduced learning times. Finally, the authors conduct experiments to exemplify the use of their algorithm. First, they investigate the influence of the tokenizer on the next token probabilities by learning surrogate automata under the top-k equivalence on $\Delta(\Sigma)$. Second, they restrict the output of a GPT-2 model numbers between 0 and 1 and learn a surrogate automata for the constraint output. They compare the output distribution of the LLM with the one from the automata and demonstrate that the automaton is able to closely approximate the original distribution.

I think the paper presents a meaningful improvement of the approach presented in [6], but I would like to criticize that the paper reads too much like an addendum to [6] in many places. This is especially true for Section 3. Without reading [6], it would not have been possible for me to comprehend this section. Another weakness is that the example use cases are arguably toy experiments. Finally, I have to point out that the submission is not according to the guidelines (missing line numbers). Because of the mentioned weaknesses, I can only give a weak recommendation for acceptance.

[6] F. Mayr, S. Yovine, M. Carrasco, F. Pan, and F. Vilensky. A congruence-based approach to active automata
learning from neural language models. In PMLR, volume 217, pages 250–264, 2023.

**Strengths:**

- Reduce number of LLM queries during learning by omitting 0-probability transitions

**Weaknesses:**

- Heavily builds upon [6]. Especially Section 3 was not really comprehensible without reading [6].
- The submission is not in the format that is demanded in the guidelines (no line number!)

**Questions:**

- Page 3 (Quotients): $\bar{\pi}([|q|]) = [\pi(q)]$ (?)
- Page 3 (Quotients): $\bar{\tau}([|q|], \sigma) = [|\tau(q, \sigma)|]$ (?)
- Page 4: What is sift? The paper would generally benefit from relying less on [6] as a reference.

---

> ### Author Response · Authors · 2024-11-20
> **Response to questions on page 3 (Quotients)**
>
> $\bar{π}([|q|])=[π(q)]$ (?)
> Yes, $π$ was missing in the RHS.
>
> $\bar{τ}([|q|],σ)=[|τ(q,σ)|]$ (?)
> Yes, it must be $q$ instead of $[|q|]$ in the RHS.
>
> We have corrected them accordingly.

---

> > ### Comment · Reviewer_gJvq · 2024-11-26
> > **Thanks for the correction**
> >
> > Thanks, that clarified thing for me. I would suggest that the authors thoroughly go through the entire formal exposition again. I think it is highly likely that I did not catch all such typos.

---

### Official Review · Reviewer_SS7n · 2024-11-04

**Soundness:** 2
**Presentation:** 1
**Contribution:** 2
**Rating:** 3
**Confidence:** 3

**Summary:**

This submission starts by presenting some background on probabilistic DFAs and describes congruences on these. The congruence of Eq (1) describes strings as equivalent if they agree on the conditional probability of each subsequent letter. The next section presents an algorithm for learning a congruence-minimal DFA based on a previous algorithm from [6]. Finally, the paper ends with a couple of experiments first looking at guiding GPT2 using an automaton, and then comparing sampling between a PDFA and GPT2.

**Strengths:**

The use of automata in the study of NN-based machine learning is extremely interesting and has led to many new insights and results (e.g. the excellent work of Weiss et al in, inter alia, [13].

**Weaknesses:**

As a general comment, this paper reads like a rushed and very early draft. I did not see a unifying story/narrative, and both motivation and details are missing

SECTION 2:

Conceptual issue: When going from eq (1) to eq (4), an equivalence relation E is introduced on the simplex \\(\Delta(\Sigma_\$)\\). Why? There's no motivation, and there are no examples. What are typical examples of equivalence relations on simplices? Whilst (1) is transparent, (4) no longer is. As a consequence, it is hard to understand where the rest of the section is going.

Technical issues:

a) The definition of \\(\mathbf{P}\\) is messy and confusing and as a consequence the proof of its existence is incomplete (and perhaps incorrect). With spaces like \\(\Sigma^\omega\\) you cannot dispense with measure theory. So the first step is to identify the measurable subsets of \\(\Sigma^\ast \cup \Sigma^\omega\\). Moreover, since you need to apply Kolmogorov's extension theorem, you in fact need to consider \\(\Sigma_\$^\omega\\). How are the measurable subsets of \\(\Sigma^\ast\\) encoded in those of \\(\Sigma_\$^\omega\\)? Next, you need to define your joint probabilities on your cylinder sets. Here you only look at very specific cylinder sets, and this is why you cannot state, let alone prove, the "permutation invariance" property required by Kolmogorov's extension theorem. That it holds is not immediately obvious to me since we're dealing with a family of conditional probabilities over a non-commutative structure (words).But since the final-dimensional distributions aren't even defined it's hard to say.

b) I'm not sure about the proof of 2.1. If we write the definition of the congruence in the better format \\(P(uw)P(v)=P(u)P(vw)\\) and we assume that \\(P(u)=0\\), then we could very well have \\(P(v)>0\\) as long as \\(P(uw)=0\\) (which seems to hold anyway by the recursive factorisation given on page 2).

c) The normalisation step is missing in the definition of \\(\mathsf{samptop}\\).

Other issues:
- The notation is sub-optimal. The subscript $ is missing at the beginning of sec 2. Why use semantics brackets [[-]] for equivalence classes?
- Fig 1: I don't see what is "troublesome" about \\(\mathcal{B}\\), it does exactly what it's supposed to be able to do if you allow non-termination.
- Either say that all proofs are in the appendix once, or state them all (there's space).
- You don't define \\(reach(Q)\\)

SECTION 3:

The reader needs some context. Why are you developing this algorithm? What is the learner? What is the teacher? What are the allowed "learning actions" (e.g. membership queries? equivalence queries? any other kind of queries?)? What is the general description of the algorithm of [6]? The reader shouldn't have to read [6] to understand what \\(\mathsf{sift}\\) is. As it stands Sec 3 cannot realistically be understood by the average reader.

The labelling of the x-axis of the LHS graph in Fig 3 is unfortunate.

SECTION 4:

This section suffers from the same problems as Sec 3. What exactly are you trying to show and why? In particular, using PDFAs seems completely overkill for the kind of simple experiments carried out in case study 1 and Table 2.
For case study 2, the guiding automaton allows every digit at ever stage, so it's not doing very much guiding. What precision of floating-point are you using? And how many digits are sampled? (the precision just gives an upper bound) The reader will not know what Outlines [14] means, this must be explained. As it stands the results for this case study mean noting to me.

- \\(\kappa\\) has not been defined anywhere
- Figs 7(a) and 7(b) are not labelled in this way.

**Questions:**

1. Should I read \\(\mathcal{L}(\sigma_1\ldots\sigma_n)\\) as \\(P[\sigma_n\mid \sigma_1\ldots\sigma_{n-1}]\\)? I don't understand if the basic model is Markovian (as the DFA model suggests) or not (as the definitions on p2 and in Prop F.1 suggest).

2. How would you define an arbitrary finite-dimensional marginal of \\(\mathbf{P}\\)?

3. What is the motivation for the \\(E\\) introduced before (3)? Can you give an example?

---

> ### Author Response · Authors · 2024-11-21
> **Response to weaknesses - Section 2 - Technical issue (a)**
>
> We politely disagree with the reviewer. In the first place, the probability measure $\mathbf{P}$ is clearly and well defined as the unique measure satisfying the conditions stated in page 2 of the paper. Its existence and uniqueness are proved in the appendix F (Prop. F.1).
>
> Regarding the measurable sets, it is standard practice to consider the sigma-algebra generated by the cylinder sets when considering probability measures over spaces of infinite sequences over finite alphabets. We will recall it at page 2 of the paper.
>
> Our proof of Prop. F.1 is based on Kolmogorov’s extension theorem, as it is stated in Theorem I.1.2 of [10] in the bibliography. For this, we first need to consider a space of infinite sequences. To achieve that, the construction embeds $\Sigma^* \cup \Sigma^{\omega}$ into $\Sigma_{\$}^{\omega}$ by adding at the end of every finite sequence in $\Sigma^*$ an infinite number of terminal symbols.
>
> This corresponds to the definition of event $A$ in the proof. Then it is enough to define a consistent family of probability measures over the cylinder sets that correspond to prefixes, as required by the cited Theorem I.1.2 of [10] (*). More precisely, for each k>=1 consider the set $Cyl(k)$, of all cylinders $C[a_1,...,a_k]$ of the form $C[a_1,...,a_k]$ = { $(x_i)_{i=1}^{\infty}: x_1=a_1,...,x_k=a_k $ }.
>
> If for each k>=1 we can define a probability $P_k$ over the cylinders $Cyl(k)$ that satisfies the consistency condition (condition (1) in page 1 of [10]: $P_k(C[a_1,...,a_k]) = \sum_{a_{k+1}} P_{k+1}(C[a_1,...,a_{k+1}]))$, then there exists a unique probability measure $\mathbf{P}$  defined on the sigma algebra generated by all cylinders of $\Sigma^{\omega}$ such that $P(C[a_1,...,a_k])=P_k(C[a_1,...,a_k])$. The final step is to prove that the event $A$ (that was previously identified with $\Sigma^* \cup \Sigma^{\omega}$) has P probability one: $\mathbf{P}(A)=1$. Uniqueness of $\mathbf{P}$ also is guaranteed by Theorem I.1.2. of [10].
>
> (*) Notice that the cited theorem does not require the consideration of (a) other cylinder sets or (b) any permutation invariance property. Indeed, for (a) under the cited consistency condition, the probability of any other cylinder set can be obtained by basic operations over sets (disjoint unions, complements, etc). For instance, if we consider two symbols a and b, and the cylinder set C=“the first symbol is a, and the third symbol is b”, then its probability can be computed as the sum of the probabilities of the cylinders C_1=“the first symbol is a, the second symbol is a, and the third symbol is b” and C_2=“the first symbol is a, the second symbol is b, and the third symbol is b”. Secondly, (b) holds automatically since the cited version of Kolmogorov’s theorem assumes that the times (positions in the sequence) are given in an increasing order so there is no need to rearrange them. This permutation invariance property is only relevant when one defines the finite dimensional probabilities as a family indexed over arbitrary finite sets of times (positions) and is not related to any underlying commutative structure. For instance, if we consider two symbols a and b, and the cylinder set C=“the first symbol is a, and the third symbol is b”, the permutation invariance property guarantees that its probability does not change if one defines it as C=“the third symbol is b, and the first symbol is a”. This is automatic if by convention one defines the probability of cylinders by considering a specific ordering of the times (increasing in our case).

---

> > ### Comment · Reviewer_SS7n · 2024-11-26
> > **Kolmogorov's extension theorem**
> >
> > I could not access [10], but standard textbooks can be consulted on this, for example
> > - Athreya, Krishna B., and Soumendra N. Lahiri. Measure theory and probability theory, see theorem 6.3.1
> > - Oksendal, Bernt. Stochastic differential equations: an introduction with applications, see theorem 2.1.5.
> >
> > Two conditions are needed to define a consistent family \\(P_t\\) of probability measures in this standard formulation:
> >
> > (i) \\( P_{t_1...t_k}(B_1\times ...\times B_{k-1}\times \mathbb{R}) = P_{t_1...t_{k-1}}(B_1\times ...\times B_{k-1}) \\)
> >
> > (ii)  \\( P_{t_1...t_k}(B_1\times ...\times B_k) = P_{t_{\pi(1)}...t_{\pi(k)}}(B_{\pi(1)}\times ...\times B_{\pi(k)}) \\)  for any permutation \\( \pi\in Sym(k)\\)
> >
> > An alternative but equivalent formulation can be found in Dudley, Richard M. Real analysis and probability, theorem 12.1.2.
> >
> > The proof in your submission only covers (i), and from the definitions it is not clear to me how (ii) could even be written down.

---

> > > ### Author Response · Authors · 2024-12-02
> > > **Follow up on Kolmogorov Extension Theorem**
> > >
> > > The relevance of condition (ii) depends on how you index the family of finite dimensional probability measures. In the versions of the theorem that you cite, it is assumed that the index $\{t_1,\ldots,t_k\}$ is a subset of positive integers, where no arbitrary specification of their order is given. In this situation it is clear that one needs the invariance of condition (ii) to ensure a well defined measure on the space of infinite sequences. The situation can be illustrated by the following: if one expects the existence of a measure $P$ over infinite sequences $\{x_t\}$ such that $P(x_{t_1}\in B_1,\ldots, x_{t_k}\in B_k)=P_{t_1\ldots t_k}(B_1\times \ldots\times B_k)$, since the intersection $\{x_{t_1}\in B_1,\ldots, x_{t_k}\in B_k\}$ is the same whatever the ordering one chooses to write it, like $\{x_{t_{\pi(1)}}\in B_{\pi(1)},\ldots, x_{t_{\pi(k)}}\in B_{\pi(k)}\}$ for a permutation $\pi$, it is a necessary condition to have (ii). But the situation is quite different if one indexes the finite dimensional probabilities $P_{t_1\ldots t_k}$ using only ordered sets of times $t_1<\ldots<t_k$. In this case it does not make any sense to write $P_{t_{\pi(1)}\ldots t_{\pi(k)}}$ for a permutation $\pi$, because unordered sets of times are not part of the index set. The passage from an indexed family $P_{t_1\ldots t_k}$ that uses only ordered sets $t_1<\ldots<t_k$ to one that uses all sets, so that the permutation invariance (ii) holds and one can use the statement of the theorem that you cite, can be done by trivially defining $P_{t_1\ldots t_k}(B_1\times \ldots\times B_k) = P_{t_{\pi(1)}\ldots t_{\pi(k)}}(B_{\pi(1)}\times \ldots\times B_{\pi(k)})$ where $\pi$ is the permutation that puts the set $\{t_1,\ldots,t_k\}$ in increasing order.

---

> > > > ### Comment · Reviewer_SS7n · 2024-12-03
> > > > **Kolmogorov Extension Theorem II**
> > > >
> > > > The standard version of Kolmogorov's extension theorem is based on an inverse system indexed by the following directed poset: (i) all finite _tuples_ of elements of a set \\(T\\) (totally ordered and thought of as time) and (ii) all injections between these (thus \\( (t_1,t_3)\leq (t_3,t_2,t_1) \\)). This explains why one needs both projections and permutations in the consistency conditions. Note that time is ordered but it makes sense to consider joint distributions indexed by any tuple, ordered or not. So this is not what distinguishes your case from this one.
> > > >
> > > > Dudley's version of the theorem, see op.cit., is based on a more compact representation using the following directed poset: (i) all finite _subsets_ of elements of a set \\(T\\) and (ii) all injections between these. It still requires a much stronger notion of consistency than your theorem.
> > > >
> > > > Your version doesn't seem to fall into any of these templates, and I'm therefore not sure calling it Kolmogorov's extension theorem is particularly useful since it conveys all the wrong intuition. If I now understand correctly, you're interested in a much simpler inverse system indexed by (i) \\(\mathbb{N}\\) and (ii) the standard ordering on \\(\mathbb{N}\\)  (i.e. \\(k\leq k+1\\)).

---

> > > > > ### Author Response · Authors · 2024-12-04
> > > > > **Follow up on Kolmogorov Extension Theorem II**
> > > > >
> > > > > Yes, we only considered the case of sequences indexed by the natural numbers with its standard ordering. The version of Kolmogorov's Extension Theorem we use is more adapted to the case when one can define the finite dimensional probabilities inductively, as it is our case with the function $P$. Other examples of standard textbooks that state the theorem in similar ways as we use it here are the following:
> > > > >
> > > > > 1. Theorem A.3.1 in Durrett, Rick. Probability: theory and examples. Vol. 49. Cambridge university press, 2019.
> > > > >
> > > > > 2. Theorem D.1 in Bass RF. Kolmogorov extension theorem. In: Stochastic Processes. Cambridge Series in Statistical and Probabilistic Mathematics. Cambridge University Press; 2011:382-384.

---

> ### Author Response · Authors · 2024-11-21
> **Response to weaknesses - Section 2 - Technical issue (b)**
>
> Definition (2) is taken from [1,11] and is given as the quotient P(uw)/P(u)=P(vw)/P(v), so zero probabilities in the denominator give undefined quotients. In the case one side of the equation is undefined the equality must be understood as implying that the other side is also undefined. In your example, u and v are not congruent according to definition (2). So, the format in which you state the equality (as a product instead of a quotient) is not equivalent to our definition (2). We will clarify this in the paper.

---

> ### Author Response · Authors · 2024-11-21
> **Response to weaknesses - Section 2 - Technical issue (c)**
>
> Thanks for the observation. We modified the paper to add the normalization.

---

> ### Author Response · Authors · 2024-11-22
> **Response to weaknesses - Section 2 - Other issues**
>
> We added the missing $ symbol at the beginning of section 2, thanks for pointing this out.
>
> The purpose of using double brackets [[.]] simple brackets [.] is to avoid confusion between congruence classes on $\Sigma^*$ (denoted with [[.]]) and equivalence classes on the simplex of probability distributions (denoted with [.]). We believe this enhances readability.
>
> The point we want to emphasize with the examples in Fig. 1 is that in order to guarantee termination when sampling from a PDFA one does not need to have positive probability of the terminal symbol $ at every state. The relevant condition is to have positive probability of  terminating in the future of every state. See “Response to question 4” to reviewer VvAN: https://openreview.net/forum?id=CIcMuee69B&noteId=GNRmkdcbz0 We will rephrase the caption of Fig. 1 to clarify this point.
>
> Please notice that $reach(Q)$ is defined where it is used in the paragraph Quotients in page 3 as the set of states reachable from the initial state following a finite sequence of transitions (defined by $\tau$).

---

> ### Author Response · Authors · 2024-11-22
> **Response to weaknesses - Section 4**
>
> - What exactly are you trying to show and why?
>
> The first goal of Sec. 4 is to formalize what it means to guide an LLM in order to get a well-defined mathematical object, in this case a PDFA, which can be further analyzed. Other works [5,14] do not provide a formalization in terms of a probabilistic automaton. The second goal is to give experimental evidence that by choosing an adequate equivalence the learned PDFA is a faithful representation of the LLM.
>
> Regarding case study 2, we agree that the use of “floating-point”, which we have taken from [14] where the case study is proposed, is misleading. Actually, the purpose is to sample numbers in the interval [0,1] written as arbitrarily long sequences of digits following an initial dot. In the first experiment, the LLM is only allowed to use digits, while in the second one, it can use all its tokens representing numbers, that totalize 994 for GPT2. In both cases, the sampling proceeds until the terminal symbol is sampled (no bound is fixed). Guiding LLM is a subject of both theoretical and practical interest and these experiments provide evidence that our approach leads to faithful representations, in the sense that the learned PDFA and the guided LLM are statistically comparable if one looks at the distribution of sampled numbers.
>
> - $\kappa$  has not been defined anywhere
>
> $\kappa$ is the parameter of the quantization as equivalence between distributions. We will explain this and provide examples of equivalences.
>
> - Figs 7(a) and 7(b) are not labelled in this way.
>
> Fig. 7(a) and 7(b) should be 7(above) and 7(below), respectively. We will put the appropriate to make this clear.
>
> - The reader will not know what Outlines [14] means, this must be explained.
>
> The appropriate bibliographic reference to Outlines has been provided. We will add a brief description of the tool to enhance readability. However, explaining the inner workings of it is out of scope of the paper.

---

> ### Author Response · Authors · 2024-11-22
> **Response to Question 1**
>
> 1. Should I read $\mathcal{L}(\sigma_1\ldots\sigma_n)$ as $P[\sigma_n | \sigma_1\ldots\sigma_{n-1}]$ ?
>
> Notice that $\mathcal{L}(\sigma_1\ldots\sigma_n)$ is a probability distribution.
>
> $\mathcal{L}(\sigma_1\ldots\sigma_{n-1})(\sigma_n)$ would be $P[\sigma_n | \sigma_1\ldots\sigma_{n-1}]$ provided $P[\sigma_1\ldots\sigma_{n-1}]$ is not zero.

---

> ### Author Response · Authors · 2024-11-22
> **Response to Question 2**
>
> 2. I don't understand if the basic model is Markovian (as the DFA model suggests) or not (as the definitions on p2 and in Prop F.1 suggest).
>
> We don’t understand what you mean by “basic model”. If one considers the sampling procedure relative to a given language model $\mathcal{L}$ defined by recursively sampling the $n$-th symbol with probability distribution $\mathcal{L}(\sigma_1\ldots\sigma_{n-1})$, provided $P[\sigma_1\ldots\sigma_{n-1}]$ is not zero, then the stochastic process $X_n$ = “the $n$-th sampled symbol” is not in general Markovian. This is even the case for PDFAs where the transition probability from one symbol to another is not well defined unless one specifies the entire past. In these cases the process could be better modeled as a Hidden Markov Process since the stochastic process Q_n=”the n-th state of the system” is indeed Markov. This does not impact the contribution of the paper regarding the active PDFA learning algorithm capable of dealing with zero probability transitions. Besides, it provides a sound basis for comparing sampling from a language model and an extracted PDFA since both processes belong to the same family of stochastic processes (i.e. Hidden Markov Models).

---

> > ### Comment · Reviewer_SS7n · 2024-11-26
> > **Conceptual confusion**
> >
> > What really confused me, and what makes this paper conceptually difficult to read is the first definition followed by the sentence
> > "Language models can be expressed in different ways, e.g., RNN, Transformers, and PDFA"
> > This is confusing because PDFAs for example span a very small subspace of what you call "language models" in the initial definition. The simple reason is that they are Markovian in the sense that the probability of each symbol in a given state is fixed and therefore does not depend on the path leading to this state. "Language models" on the other hand allow completely arbitrary history dependencies. Yet PDFAs and "language models" are discussed simultaneously throughout the paper, which makes it very difficult to follow.
> >
> > To add to this confusion, it is not clear to me which "language models" correspond to probability measures constructed by Kolmogorov extension. It is only for these that the discussion on, e.g., congruences applies.

---

> > > ### Author Response · Authors · 2024-12-02
> > > **Response to "Conceptual confusion"**
> > >
> > > Comment:
> > >
> > > What really confused me, and what makes this paper conceptually difficult to read is the first definition followed by the sentence "Language models can be expressed in different ways, e.g., RNN, Transformers, and PDFA" This is confusing because PDFAs for example span a very small subspace of what you call "language models" in the initial definition. The simple reason is that they are Markovian in the sense that the probability of each symbol in a given state is fixed and therefore does not depend on the path leading to this state. "Language models" on the other hand allow completely arbitrary history dependencies. Yet PDFAs and "language models" are discussed simultaneously throughout the paper, which makes it very difficult to follow.
> > >
> > > Response:
> > >
> > > Let $\mathcal{L}$ be any language model and take the congruence $\equiv^\bullet$  defined in Eq. (6) with \(E\)  the equality of distributions. The set of congruence classes $ [[\Sigma^\ast]] $ is the state space of $\mathcal{L}$, $ \tau([[u]],\sigma) = [[ u \sigma ]] $  is well defined, as well as $ \pi([[u]]) $, because all $v$ congruent with $u$  has the same probability distribution (since $E$  is equality). There is no difference between $\mathcal{L}$  and $ ([[\Sigma^\ast]], [[\lambda]], \tau, \pi)$  since $\mathcal{L}(u) = \pi^\ast (u)$  for every $u$ (that is $\mathcal{L}$ and $\pi^\ast$ are the same function). The latter is a probabilistic deterministic automaton with possibly infinite states. Actually, if $ [[\Sigma^{*}]] $  is finite, it is a PDFA. As we have already mentioned in the previous response to question 2, the Markovian property (in the sense you mentioned) for $\mathcal{L}$  holds if you look at the sequence of states which is completely determined by the sequence of symbols. So, the only possible difference between the general definition of language model (Def. 1) and that of PDFA is the finiteness of the set of states.
> > >
> > > Comment:
> > >
> > > To add to this confusion, it is not clear to me which "language models" correspond to probability measures constructed by Kolmogorov extension. It is only for these that the discussion on, e.g., congruences applies.
> > >
> > > Response:
> > >
> > > Proposition 2.1 (which relies on Kolmogorov extension theorem) applies to any language model as defined in Definition 1. The definition of congruence does not rely neither on Prop. 2.1 nor on Kolmogorov extension theorem. It only depends on $P$ (not on $\mathbf{P}$).

---

> ### Author Response · Authors · 2024-11-22
> **Response to Question 3**
>
> 3. How would you define an arbitrary finite-dimensional marginal of $\mathbf{P}$?
>
> For instance, if we consider an alphabet of two symbols $a$ and $b$, and the cylinder set $C$ = “the first symbol is $a$, and the third symbol is $b$”, then its probability can be computed as the sum of the probabilities of the cylinders $C_1$ = “the first symbol is $a$, the second symbol is $a$, and the third symbol is $b$” and $C_2$ = “the first symbol is $a$, the second symbol is $b$, and the third symbol is $b$”.

---

> ### Author Response · Authors · 2024-11-22
> **Response to Question 4**
>
> 4. What is the motivation for the $E$ introduced before (3)? Can you give an example?
>
> Resorting to some kind of tolerance relation between distributions is usual practice when it comes to approximating the behavior of language models with probabilistic automata (e.g. [Weiss et al 2019, Clark and Thollard, 2004]), in order to group in a single state strings which continuations slightly differ in probability. Eventually, this grouping could result in an approximation with a finite number of states even if the image of the language model contains infinitely many distributions, while keeping the error of the approximation as small as desired or preserving the property to be checked. Moreover, using equivalences instead of tolerances (e.g., reference [6] in the paper) leads to a well-defined notion of algebraic quotient and allows capturing the behavior of the language model under usual sampling strategies such as top-k through the associated top-k equivalence, defined as two distributions are top-k equivalent if their k most probable symbols are the same, or obtaining probabilistic automata which are consistent with specific performance metrics such as word error rate (WER) or normalized discounted cumulative gain (NDCG). In the paper we used quantization equivalence: for instance, for $\kappa=2$ we have the partition of the interval $[0 \ldots 1]$ into the set of quantization intervals $[0]$, $(0,0.5)$, $[0.5,1)$, $[1]$, and two distributions $\delta_1$ and $\delta_2$ are equivalent if for each symbol $\sigma$, $\delta_1(\sigma)$ and $\delta_2(\sigma)$ fall into the same quantization interval. In particular, the singleton intervals $[0]$ and $[1]$ are needed to individualize 0 probabilities. Of course, other variants could be defined. We will clarify this and add the appropriate references.
>
> References:
>
> [Weiss, Goldberg, and Yahav, 2019] Learning Deterministic Weighted Automata with Queries and Counterexamples. NeurIPS, 2019.
>
> [Clark & Thollard, 2004] Alexander Clark and Franck Thollard. Pac-learnability of probabilistic deterministic finite state automata. Journal of Machine Learning Research, 5:473–497, 2004.

---

### Official Review · Reviewer_VvAN · 2024-11-05

**Soundness:** 3
**Presentation:** 1
**Contribution:** 2
**Rating:** 3
**Confidence:** 4

**Summary:**

# Summary
The paper addresses the problem of language models, where specific symbols during text generation may have a probability of zero. The authors tackle the problem by formalizing the problem in an automata-theoretic setting and defining a congruence relation on sequences that depends on (1) the extension of sequences (as is usual) and (2) the occurrence of symbols with zero probability. Based on the congruence relation, the authors propose an active automata learning called Omit-Zero. They evaluate the runtime of the algorithm in comparison to an existing algorithm called QNT. Finally, they apply the algorithm in case studies where the symbol generation of language models is guided.

**Strengths:**

+ Systematic analysis of LLMs is a timely topic, and a solid theoretical foundation can benefit the community.
+ The algorithm and the claims seem generally sound.

**Weaknesses:**

+ The presentation of the paper needs to be significantly improved.
    + There is a lot of notation and terminology, which is not bad per se, but some parts need to be explained better, and it might be possible to skip some parts. For example, the term "quotient" is used already in the second sentence of the abstract, but only on Page 3 (bottom), it becomes clear what a quotient looks like.
    + Section 3 can be understood with some knowledge of automata learning, however, I assume that most readers don't immediately know what "sift" means. The relevance of some concepts, like the Hopcroft-Karp algorithm, is unclear.
It seems to me that the paper could benefit from a more stringent focus on the most relevant concept while abstracting away some details.
+ It is unclear how the case studies relate to the main problem considered by the paper.
    + Case Study 1: Here, we see a difference between different tokenizer settings, but it is unclear what the effect of the 0-probabilities problem would have been.
    + Case Study 2: This case study examines the fidelity of learned automata, but the problem and its (potential) effect are not mentioned.
From the introduction, I would have assumed to see an experiment showing non-termination or some similar effect.

**Questions:**

1. Would the case study have been possible with QNT?
2. Does Corollary 2.1 hold for all PDFA, or does it require minimality?
3. In the runtime experiment, the probability of a symbol to be 0 is at least 0.9. Is this probability realistic when considering the guiding of an LLM?
4. Considering the problem of non-termination illustrated in the caption of Fig.1. It seems to me that a simple solution would be to avoid excluding $ using top_r. Am I missing something?

---

> ### Author Response · Authors · 2024-11-19
> **Response to question 2**
>
> 2. Does Corollary 2.1 hold for all PDFA, or does it require minimality?
>
> Corollary 2.1 holds for every language model, therefore including all PDFA. Actually, notice that #[[ \Sigma* ]] is the number of congruence classes over \Sigma* induced by the language model. For a PDFA A #[[ \Sigma* ]] turns out to be the number of states of the quotient \bar{A} which is the smallest PDFA congruent with A (see Quotients paragraph right below Corollary 2.1). That is, A and \bar{A} define the same congruence. Besides, #[[ \Sigma* ]] may be significantly smaller than the upper bound given by Corollary 2.1 when the percentage of 0-probability transitions increases. We will add a figure together with the experiments in Section 3 to illustrate this.

---

> > ### Comment · Reviewer_VvAN · 2024-11-22
> > **question 2 clarification**
> >
> > Sorry, I confused Corollary 2.1. with Equation (6). I meant to ask if Equation (6) and rephrasing \equiv_E over E require minimality.

---

> > > ### Author Response · Authors · 2024-11-23
> > > **Follow up on question 2**
> > >
> > > Minimality is not required.
> > >
> > > Equation (6) defines the congruence over states via the congruence over strings: it says that two states are congruent iff they are reached by congruent strings are congruent. You could have defined congruence over states otherwise: two states $q$ and $q’$ are congruent iff for every continuation $w$, $[\pi^\ast(q, w)] = [\pi^\ast(q’,w)]$. Then, it follows that if $q=\tau^\ast(u)$ and $q’=\tau^\ast(v)$ are congruent then $u$ and $v$ will be congruent as well since $[\pi^\ast(uw)] = [\pi^\ast(q, w)] = [\pi^\ast(q’,w)] = [\pi^\ast(vw)]$ for all $w$. The reverse also holds: take $u$ and $v$ congruent, then for all $w$, $[\pi^\ast(uw)] = [\pi^\ast(vw)]$, and therefore $[\pi^\ast(q, w)] = [\pi^\ast(q’,w)]$, so $q$ and $q’$ are congruent. Hence, both definitions are equivalent.
> > >
> > > A PDFA is minimal iff for any pair of distinct states $q$ and $q’$ it happens that $q$ and $q’$ are not congruent. This implies that if $q=\tau^\ast(u)$ and $q’=\tau^\ast(v)$, with $q$ different from $q’$, then $u$ and $v$ are not congruent either. And conversely. Overall this implies that a minimal PDFA has as many states as the quotient defined by the congruence over $\Sigma^\ast$.

---

> ### Author Response · Authors · 2024-11-19
> **Response to question 3**
>
> 3. In the runtime experiment, the probability of a symbol to be 0 is at least 0.9. Is this probability realistic when considering the guiding of an LLM?
>
> Indeed, the probability of 0 is likely to be grater than 0.9 in reality. For instance, Kuchnik et al [5] cited in the bibliography, uses top-40 for analyzing memorization and toxicity (also citing other papers) in the case of GPT-2 which has 50257 tokens, thus giving 0.999 probability of a symbol to have a probability of 0 under this sampling strategy. The same paper uses k=1000 for language understanding, which yields 0.98, and acknowledges that it is a conservative span. For GPT-2, 0.9 would consist in sampling a set of approximately 5000 tokens, which is typically too large according to the literature. Therefore, it would not be realistic to test the algorithm on smaller probabilities.

---

> > ### Comment · Reviewer_VvAN · 2024-11-22
> > **question 3 follow up**
> >
> > Okay, I understand that, but your experiments use |\Sigma|=20 instead of 50257. Does this not change the setting drastically? With top-40, I have 40 symbols with non-zero probability.

---

> > > ### Author Response · Authors · 2024-11-28
> > > **response to question 3 follow up and question 1**
> > >
> > > We added experiments that provide empirical evidence that Omit-Zero is significantly faster than Teacher-Filter (that is QNT with the help of a filter of 0-probability transitions) in case study 2 when 994 numeric tokens are used. Figure 6 in the revised version summarizes these results.

---

> ### Author Response · Authors · 2024-11-19
> **Response to question 4**
>
> 4. Considering the problem of non-termination illustrated in the caption of Fig.1. It seems to me that a simple solution would be to avoid excluding $ using top_r. Am I missing something?
>
> In general, it may not be appropriate to be able to terminate at every state. Take for example case study 2. The guide does not allow to generate the string "." since it is not a legal number. Allowing $ to be in the support of the distribution of state q1 (Fig. 7(b)) would (wrongly) allow this string to be generated. Something similar happens in case study 1, where it is required to complete a full sentence.
>
> Therefore, always having $ in the support of the distribution would prevent capturing these examples.
>
> The desired property could be stated as follows: the probability of eventually terminating is equal to 1. This corresponds to P_$ (defined in pg. 2) to be a probability distribution.
>
> Indeed, this property can be expressed in the probabilistic temporal logic PCTL [PMC] as the formula P(<>p)=1, where p is a proposition that is true at a state q if and only if $ is in the support of the π(q). For instance, such formula could be model checked on a probabilistic automaton with the tool PRISM [PRISM].
>
> References:
>
> [PMC] Baier, Ch. and Katoen, J.P. Principles of Model Checking. MIT Press, 2008.
>
> [PRISM] https://www.prismmodelchecker.org/

---

> > ### Comment · Reviewer_VvAN · 2024-11-22
> > **Question 4 follow up**
> >
> > This was not what I meant. I mean if $ is in the support before applying top_r it should be in the support afterwards. If it is not in the support before, I would not want to add it.

---

> > > ### Author Response · Authors · 2024-11-23
> > > **Follow up on question 4**
> > >
> > > Where to enable termination depends on the application. Our approach does not force any particular behavior.
> > >
> > > Of course, you can define say a sampling strategy top'_r as you suggest which will let the terminal symbol to remain in the support of the resulting distribution when it is permitted by the guide (since synchronization with the guide is done before applying the sampling strategy) if this is the behavior you want.
> > >
> > > Now, using top'_ r or top_r may result in different languages because the set of strings for which P_$ is not null may be different. In any case, learning a PDFA will also help checking and understanding that.

---

> > > > ### Comment · Reviewer_VvAN · 2024-12-03
> > > > **PDFA helps understanding**
> > > >
> > > > Thanks, I can see the value of PDFA helping understanding.

---

### Author Response · Authors · 2024-11-19
**Comments on Section 3**

We would like to thank the reviewers for the detailed comments. All reviews have pointed out that Section 3 lacks sufficient detail to understand how the algorithm works as it depends on results on the literature and on context on automata learning. We will write a general response and rewrite Section 3 taking into account the comments.

---

> ### Author Response · Authors · 2024-11-23
> **Comments about Section 3 (continued) - Overview of the algorithm**
>
> Omit-Zero (as well as QNT) is inspired by Kearns and Vazirani’s algorithm [KV].
>
> Omit-Zero keeps a search tree whose leafs (Acc) represent congruence classes and whose inner nodes (Dis) are such that the least common ancestor of two leafs is a string showing that the leafs are actually not congruent. That is why inner nodes are called distinguishing, because they serve as evidence to disprove congruence.
>
> Sift is the search procedure: given a string, it finds the congruence class (leaf) where it possibly belongs to. If no such leaf exists, it means that a new class has been found and it is added as a leaf to the tree by sift-update. Sift is used by the procedure build to construct an automaton from the tree: states are leafs and transitions from one state to another are found using sift (concatenating the string representing the congruence class, so called access string, with all symbols in the support of the leaf). Once the automaton is built, the algorithm checks if it is congruent with the target language model. If they are not, the counterexample is evidence of the existence of a class that is not in the tree. Then, procedure update adds a new leaf to the tree. The algorithm starts with an initial tree containing only the root (the empty string) from which a first automaton is built by InitilizeHypothesis.
>
> EQ is the equivalence query, which is the procedure responsible for checking whether the hypothesis and the target are congruent. When the target is an automaton, EQ can be implemented by the Hopcroft-Karp algorithm for testing equivalence of finite automata (reference [3] in the bibliography). We used an adaptation of this algorithm to perform the experimental evaluation of Omit-Zero in Section 3, because it allows to efficiently check equivalence between the obtained PDFA and the target PDFA. Now, when the target system involves an LLM which is not an automaton, it is no longer possible to use it. In this case, it is standard to resort to sampling. In order to ensure that every sampled string $u$ is defined, that is, $P(u)>0$, we sample from the hypothesis PDFA using random walk (which is sound because of Proposition 2.4).
>
> Sifting a string $v$ defines a path which is the sequence of distinguishing strings (inner nodes) traversed by the sift operation when processing $v$ from the root to the leaf. In order to ensure that an inner node is indeed a valid evidence of non-congruence, it must have a defined prefix (Proposition 2.4). This is guaranteed by requiring that every inner node starts with a symbol in the support of the associated distribution (equation 8). Such requirement is fulfilled jointly by the correct processing of the counterexample which finds a defined prefix of it and procedure update when it adds an inner node. Then, every inner node $w$ in the path followed by sift for a string $v$ to leaf $u$ is a correct evidence that $v$ could not possibly belong to any other equivalence class in the tree different from $u$.
>
> References:
>
> [KV] Kearns, M., and Vazirani, U. V. (1994). An Introduction to Computational Learning Theory.

---

### Author Response · Authors · 2024-11-28
**Revised version**

We uploaded a revised version of the paper that we hope addresses the weaknesses and questions raised by the reviewers.

---

### Meta-Review · Area_Chair_MYd2 · 2024-12-20

**Metareview:**

This paper uses a word congruence for a language model. The goal is to learn (surrogate) probabilistic automata using a typical Myhill-Nerode theorem. Particularly, the problem of sequence with probability 0 is tackled.

All reviewers agree that the paper requires further polishing and clarification, and therefore, another round of peer-review is required.

**Additional Comments On Reviewer Discussion:**

The reviewers acknowledge that the paper has improved during the discussion phase, but still an improvement in writing and presentation is required.

---

### Decision · Program_Chairs · 2025-01-22

Reject